# Multi-level and lineage-specific interactomes of the Hox transcription factor Ubx contribute to its functional specificity

Julie Carnesecchi [1], Gianluca Sigismondo[2,3,4], Katrin Domsch[1,4], Clara Eva Paula Baader[1], Mahmoud-Reza Rafiee[2], Jeroen Krijgsveld[2,3] & Ingrid Lohmann [1✉]

Transcription factors (TFs) control cell fates by precisely orchestrating gene expression. However, how individual TFs promote transcriptional diversity remains unclear. Here, we use the Hox TF Ultrabithorax (Ubx) as a model to explore how a single TF specifies multiple cell types. Using proximity-dependent Biotin IDentification in *Drosophila*, we identify Ubx interactomes in three embryonic tissues. We find that Ubx interacts with largely non-overlapping sets of proteins with few having tissue-specific RNA expression. Instead most interactors are active in many cell types, controlling gene expression from chromatin regulation to the initiation of translation. Genetic interaction assays in vivo confirm that they act strictly lineage- and process-specific. Thus, functional specificity of Ubx seems to play out at several regulatory levels and to result from the controlled restriction of the interaction potential by the cellular environment. Thereby, it challenges long-standing assumptions such as differential RNA expression as determinant for protein complexes.

[1] Heidelberg University, Centre for Organismal Studies (COS) Heidelberg, Department of Developmental Biology and CellNetworks - Cluster of Excellence, Heidelberg, Germany. [2] German Cancer Research Center (DKFZ), Im Neuenheimer Feld 581, 69120 Heidelberg, Germany. [3] CellNetworks - Cluster of Excellence, and Heidelberg University, Medical Faculty, Im Neuenheimer Feld 672, 69120 Heidelberg, Germany. [4] These authors contributed equally: Gianluca Sigismondo, Katrin Domsch. ✉email: ingrid.lohmann@cos.uni-heidelberg.de

The development of living organisms is the result of a fine-tuned spatial and temporal expression of genes, which is driven by transcription factors (TFs). Many TFs are expressed in several cell types, and control different transcriptional programs depending on the cell context[1–4]. However, how multi-lineage TFs can function in such specific manner in different environments remains elusive. Most of the efforts to understand the function and specificity of TFs was so far focused on their interaction with regulatory proteins at *cis*-regulatory modules, so-called enhancers and promoters[5–8]. However, TFs do not only interact with other TFs but with a variety of proteins including chromatin associated proteins, histone modifiers, factors of the general transcriptional machinery or mRNA regulatory proteins[9–13]. Hence, it is thought that TFs promote cell type diversity by assembling protein interaction networks consisting of different types of proteins in a cell-type-specific manner[6,14,15]. However, as suitable approaches have been unavailable so far, this assumption still awaits approval.

One prominent example of broadly expressed TFs is the conserved class of Hox proteins, which are active in many embryonic and adult tissues along the anterior-posterior (A/P) axis of animals[16]. Although Hox TFs recognize similar DNA sequences in vitro due to a highly conserved DNA-binding domain, the homeodomain (HD)[17], they control gene expression programs in a highly context-dependent manner in vivo via the interaction with other proteins[2,18,19]. In particular, the interaction with the three-amino acids loop extension (TALE) family of HD-containing TFs has been extensively studied, which includes the *Drosophila* Extradenticle (Exd) and the vertebrate Pbx1-4 proteins[20]. These proteins cooperatively bind DNA with Hox TFs thereby increasing their regulatory specificity[20–23]. Hox-TALE interactions are mostly mediated via a short hexapeptide (HX) motif, which lies upstream of the Hox HD[24], and alternatively via the UbdA domain, a protein motif found downstream of the HD in the two Hox TFs Ultrabithorax (Ubx) and Abdominal-A (Abd-A)[25,26]. Although TALE TFs are important for Hox function, they can only partially explain how Hox TFs can function in a context-specific manner in vivo, in particular as they are expressed in many different cell types themselves[27]. Thus, Hox proteins are an ideal model to tackle the question of how TFs orchestrate precise transcriptional programs in different cellular contexts.

In order to reveal the regulatory complexes that drive the multi-faceted outputs of TFs, unbiased methods are required to identify stable and transient TF interaction networks in vivo. Proximity-labelling of proteins coupled with mass spectrometry (MS) offers a systematic analysis of spatially restricted proteomes, providing a comprehensive understanding of cellular functions in different contexts[28–32]. The two most prominent proximity-labelling methods are Ascorbate peroxidase proximity labelling (APEX) and proximity-dependent biotin identification (BioID), which are both based on biotinylation of adjacent proteins followed by affinity-based purification[29,32,33]. Thus, these two methods allow capturing and identifying the neighbourhood proteins in the context of a living cell. In contrast to APEX, BioID, whose activity depends on biotin, does not alter cell physiology[29,34]. In this system, the close-proximity biotinylation is driven by a mutant version of the biotin-ligase BirA originating from *Escherichia coli*. This mutant version called BirA* (R118G) converts biotin into the reactive compound 5′-bioAMP but loses its affinity for this substrate. BioAMP is then released and biotinylates proteins on lysine residue in a 10 nm range[29,34,35]. BioID has been applied in multiple systems ranging from cell culture to tumour xenografts in mice[29,36,37].

Here, we combine BioID with the GAL4-UAS system[38], which permits the expression of the BirA* fusion protein in the cell type of choice and allows to capture lineage-specific interactomes. We use the Hox TF Ubx as a model, as it specifies distinct developmental programs in different tissue types in a stage-dependent manner[2]. For our comparative analysis of Ubx interactomes, we focus on the mesodermal, neural and neuroectodermal lineages. Our results demonstrate that targeted BioID is highly efficient in isolating lineage-specific Ubx partners at the subcellular level in vivo, and reveal that Ubx interactomes in the different lineages were largely non-overlapping. Interestingly, we find that Ubx interacts mostly in a lineage-specific manner with ubiquitously expressed proteins involved in general transcriptional regulation, like chromatin remodelling proteins or RNA processing factors, and only with a few of lineage-restricted factors. Even more important, our genetic interaction analyses reveal that, in vivo, the identified interactions acted lineage- and process-specifically. It demonstrates that functional specificity of Ubx is realized at multiple regulatory levels and is not only a consequence of different Ubx-protein combinations recognizing distinct sequence codes written in enhancers and promoters. Thus, TFs seem to act as versatile protein platforms, which function beyond the *cis*-regulatory level to ensure robust yet flexible gene expression programs critical for the development and maintenance of cell and tissue types.

## Results

**Design and validation of BioID in a *Drosophila* cell system**. To identify lineage-specific interaction partners of the Hox TF Ubx in vivo, we combined BioID with the GAL4-UAS system[38]. To this end, we fused the N-terminal part of Ubx (isoform Ia) to UAS-myc-BirA* (mB*Ubx$^{WT}$) (see Methods) (Fig. 1a). In addition, we also generated a fusion of BirA* and Ubx containing a single mutation (N51A) in the DNA-binding domain, the homeodomain (mB*Ubx$^{N51A}$). This mutation prevents the recognition and binding of Ubx to DNA, which we confirmed by electrophoretic mobility shift assay (EMSA) (Supplementary Fig. 1a). We reasoned that a comparison of Ubx$^{WT}$ and Ubx$^{N51A}$ interactomes would allow the discrimination of interactions important for TF binding to the chromatin from interactions established in the nucleoplasm (Fig. 1b). As a general control, BirA* was fused to GFP and a nuclear localisation sequence (mB*nlsGFP). In order to verify the suitability of BioID for identifying Ubx interaction partners, we tested the system in Drosophila S2R+ cells (see Supplementary Note 1, Supplementary Fig. 1).

Taken together, these results demonstrated that BioID is an efficient and specific method to purify interaction partners of TFs in a *Drosophila* cell-based system.

**Establishment of targeted BioID in *Drosophila* embryos**. Having confirmed the efficiency of BirA*Ubx fusion proteins in biotinylating close-proximity proteins in cells, we next tested the technique in embryos and generated transgenic flies carrying the mB*Ubx$^{WT}$, mB*Ubx$^{N51A}$ and mB*nlsGFP fusions. First, we verified the functionality of the proteins in living animal by analysing the well-described homeotic transformation induced by aberrant Hox expression[39] and used the transformation of segmental denticle belt patterns in first instar larvae as a read-out. In line with previous reports[1], ubiquitous expression of wild-type Ubx (mB*Ubx$^{WT}$) induced a switch of thoracic segment identity towards the identity of abdominal segments but not mB*Ubx$^{N51A}$ (Fig. 1d). These results verified that the mB* fusion proteins are functional in *Drosophila*.

To resolve cell type-specific Ubx interactive networks, we selected the mesodermal and neural tissues due to the well-described function of Ubx in both lineages[2,40]. Using the pan-

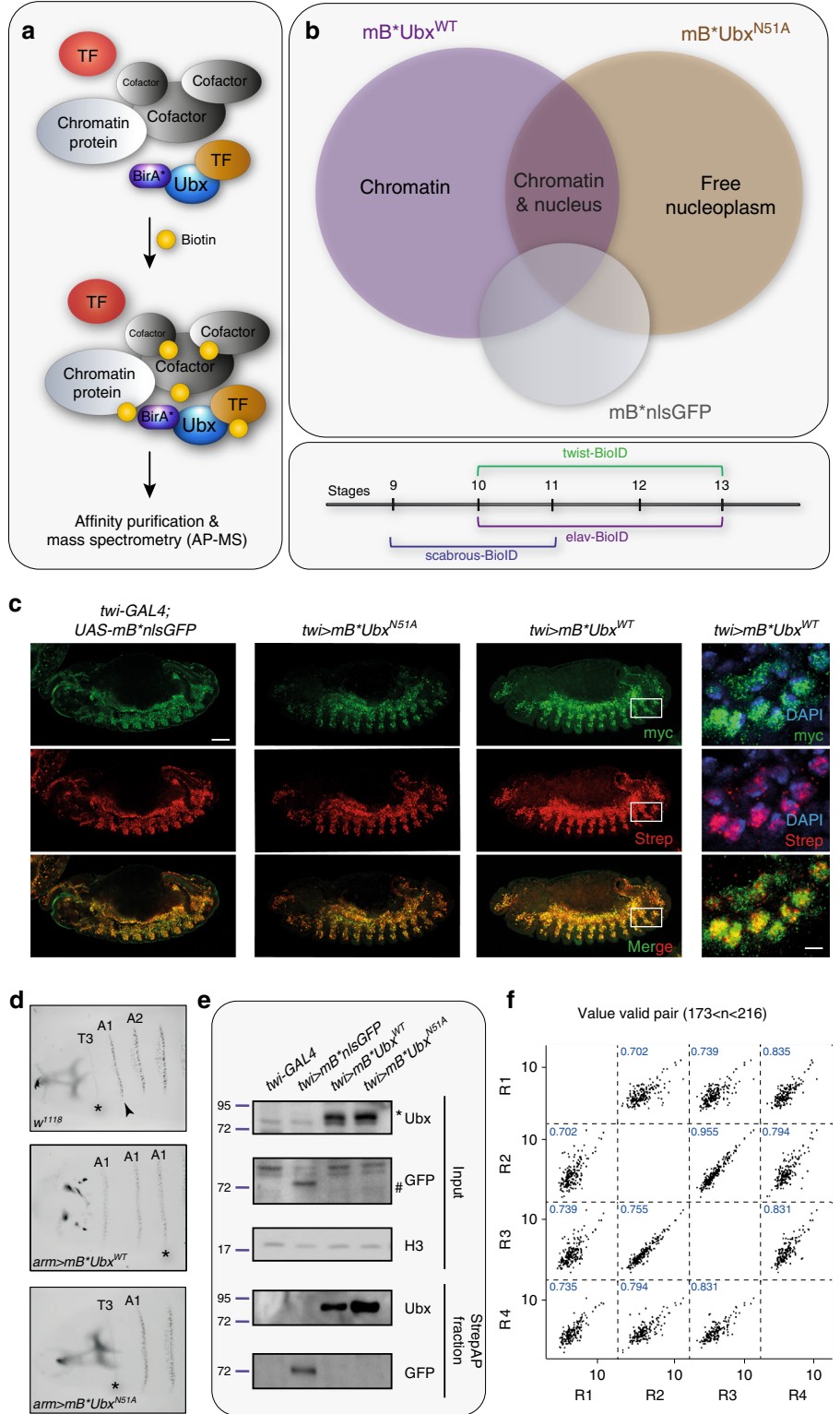

mesodermal driver *twist*-GAL4 (*twi*-GAL4) and the pan-neural driver *elav*-GAL4, we expressed the mB* fusion proteins in stage 10–13 embryos (5–8 h after egg lay AEL) (Fig. 1b). We selected this time frame as Ubx is normally expressed and active in these tissues during these stages[2]. To control for any discrepancies in lineage-specific timing, we also mapped the Ubx interactome in the early nervous system (stage 9–11 embryos, 2.5–5 h AEL) using the neuroectodermal driver *scabrous*-GAL4 (*sca*-GAL4) (Fig. 1b).

We first evaluated the tissue-specific expression of the mB* fusion proteins and their activity by immunofluorescence. This analysis revealed a robust and specific expression and biotinylation efficiency of the BirA* fusion proteins (Fig. 1c, Supplementary Figs. 2b–d, 3a, b). In contrast, we did not detect any biotinylation in wild-type embryos (Supplementary Fig. 2a). These results demonstrated that the yeast-rich food diet used for the experiments was sufficient for BirA* dependent protein biotinylation in *Drosophila* embryos, rendering biotin

**Fig. 1 Design of targeted BioID in *Drosophila* embryos. a** Representation of BioID-MS. In the presence of biotin, proteins in close proximity to the BirA*-Ubx protein are biotinylated and subjected to mass spectrometry (MS) upon affinity purification. **b** Top panel: BioID design to identify interactions occurring on the chromatin fraction and in the nucleoplasm. Close-proximity partners identified with Ubx wild-type version (purple) were compared with close-proximity partners identified with an Ubx mutant version (brown), not able to bind DNA. The BirA* protein fused to nlsGFP (grey) was used as a control. Bottom panel: Design of embryo collections for BioID performed in the three tissues: mesoderm (twist-BioID), nervous system (elav-BioID) and neuroectoderm (scabrous-BioID). The numbers correspond to embryonic stages. **c** Immunostaining of stage 13 embryos (5–8 h AEL), expressing *UAS-mB*nlsGFP* (left), *UAS-mB*Ubx^N51A* (left-middle) or *UAS-mB*Ubx^WT* (right-middle) transgenes in the mesoderm by means of the *twi-GAL4* driver. Transgene expression is shown by myc (green), biotinylated proteins by streptavidin (red) stainings, merge highlights specificity of biotinylation. Right panel: close-up of nuclei (white boxes in the right-middle panels), DAPI (blue) marks the DNA. **d** The anterior part of cuticles of *w^1118*, *arm>mB*Ubx^WT* and *arm>mB*Ubx^N51A* 1st instar larvae are shown. Ubiquitous overexpression of the BirA*-Ubx^WT fusion protein resulted in the transformation of T3 (asterisk) and more anterior segments into the identity of A1 (black arrowhead), and the head skeleton was malformed. Overexpression of the BirA*-Ubx^N51A protein had no effect on segment identity. **e** Western blots of streptavidin affinity purification (BioID) performed on extracts of *twi-GAL4*, *twi>mB*nlsGFP*, *twi>mB*Ubx^WT* and *twi>mB*Ubx^N51A* embryos. Input and streptavidin purified fraction (StrepAP) are shown. Ubx, GFP, histone H3 (H3) antibodies were used for detection. The asterisk indicates Ubx, the rhomb GFP proteins. Protein size is indicated relative to ladder position. **f** Pearson correlation (blue) of four replicates of BioID samples retrieved from twi-Ubx^WT embryos indicated a high correlation between the replicates. The number of valid proteins used for subsequent analysis is indicated. Images are representative of all embryos analysed per genotype over 3 sets of embryos collection from independent crossings. Scale bar: 50 μm; zoom scale bar: 5 μm. See also Supplementary Figs. 1–3 and Supplementary Data 1–33. Source data are provided as a Source Data file.

supplementation unnecessary in vivo. Detailed analysis of BirA* fusion protein expression and biotinylation confirmed the specificity of the system, as both BirA* expression and biotinylation were exclusively detected in the lineage and at the time-points controlled by the different drivers (Fig. 1c, Supplementary Figs. 2b–d). Finally, western blot analysis revealed an efficient streptavidin affinity purification of biotinylated proteins using nuclear extracts from *twi>mB*Ubx^WT*, *twi>mB*Ubx^N51A* and *twi>mB*nlsGFP* embryos (Fig. 1e).

In sum, these results showed that the targeted BioID method is efficient and highly specific in embryos and thus ideally suited to study spatiotemporal interactomes of Ubx.

**Exploring targeted BioID in *Drosophila* embryos.** We subsequently performed mass spectrometry analysis using the streptavidin affinity purified fraction of nuclear extracts from embryos expressing the BirA* fusion proteins (mB*Ubx^WT, mB*Ubx^N51A and mB*nlsGFP) under the control of the *twi-*, *elav-* and *sca-*GAL4 drivers. The experiments had high similarities across independent biological replicates for both the neural and mesodermal BioID (Pearson correlation, $n = 4$; twi-BioID $r > 0.7$; elav-BioID $r > 0.85$) (Fig. 1f, Supplementary Figs. 3c, e). In contrast, replicates of the neuroectodermal BioID were more variable ($r > 0.58$, Supplementary Fig. 3d), which may be a consequence of the broad activity of the *sca-*GAL4 driver in a mixed cell population consisting of ectodermal and neural progenitor cells[41]. The origin of the GAL4 driver also controlled the amount of proteins detected by BioID. For example, the total number of proteins quantified was between 142 and 244 for the mesoderm (Fig. 1f, Supplementary Fig. 3e), between 70 and 131 for the neural system and 242–593 for the neuroectoderm (Supplementary Fig. 3c, d). This discrepancy is likely due to the different activities of the *elav-* and *twi-*GAL4 drivers[2], resulting in a shorter biotinylation period in the elav-BioID sample (Supplementary Fig. 2b, c), while the *sca-*GAL4 targets more cells in comparison to the *twi-* and *elav-*GAL4 drivers (Supplementary Figs. 2b, d, 3b), allowing more proteins to be biotinylated.

In order to identify features characterizing the different Ubx BioID-interactomes, we performed principal component analysis (PCA) as well as heat map representations on all of the proteins found in Ubx^WT replicates from the different tissues. We specifically used the proteins of the Ubx^WT datasets, as they included Ubx interactions normally established in the different tissues. Both approaches grouped replicates of Ubx BioID-interactomes based on the lineage identity (Fig. 2a,

Supplementary Fig. 4a), showing that the lineage context dictated the interaction partners of Ubx. We next compared the different datasets using Pearson correlation coefficient analysis. We found that the mesodermal (twi-BioID) and neural (elav-BioID) Ubx BioID-interactomes were the most similar datasets ($r = 0.66$ for twi-/elav-BioID), while the neuroectodermal (sca-BioID) and mesodermal Ubx BioID-interactomes showed the greatest differences ($r = 0.245$ for sca-/twi-BioID) (Supplementary Fig. 4b). This result highlighted once more the importance of the lineage context but also showed that Ubx interactions are dependent on the developmental stage.

In sum, targeted BioID allowed us to identify lineage- and stage-specific Ubx interactomes, which we assumed to be at the basis of Ubx's ability to orchestrate functional diversity during development by triggering distinct and highly defined gene expression programs in a spatial and temporal manner.

**Characterization of lineage-specific Ubx BioID-interactomes.** We next analysed the proteins that were found in the vicinity of Ubx in the mesodermal, neural and neuroectodermal lineages. To this end we compared proteins which were significantly enriched in the Ubx^WT samples by normalising them to the GFP control and selected the ones enriched in 2 out of 4 replicates (see Methods, Supplementary Data 4–33 and Supplementary Table 1). This analysis resulted in the recovery of 60 proteins specific for the mesoderm, 19 for the nervous system and 78 for the neuroectoderm (Fig. 2b). Intriguingly, the vast majority of proteins was unique for each Ubx BioID-interactome (135/145), while only 10 were found in more than one BioID-interactome with two of them, Ubx itself and Brahma associated protein 111kD (Bap111, Dalao) a component of the Brahma nucleosome remodelling complex, identified as Ubx close-proximity partners in all tissues (Fig. 2b, d, Supplementary Data 34). This result raised the question whether these differences in Ubx interactomes are a consequence of the interactors being differentially expressed in the individual cell types. To test this, we analysed the expression of Ubx^WT close-proximity partners using lineage- and stage-specific transcriptome data[2], and found that the majority of Ubx BioID partners were equally expressed in the mesoderm and nervous system (Supplementary Data 35). Only a few BioID hits showed tissue-specific expression, which included two out of 60 proteins in the mesoderm (Tinman, Tin and Brick a brac 2, Bab2) and two out of 19 proteins in the neural system (TfAP-2 and Grainy-head Grh). This result demonstrated that although most of the Ubx interactors were broadly expressed, Ubx was able to

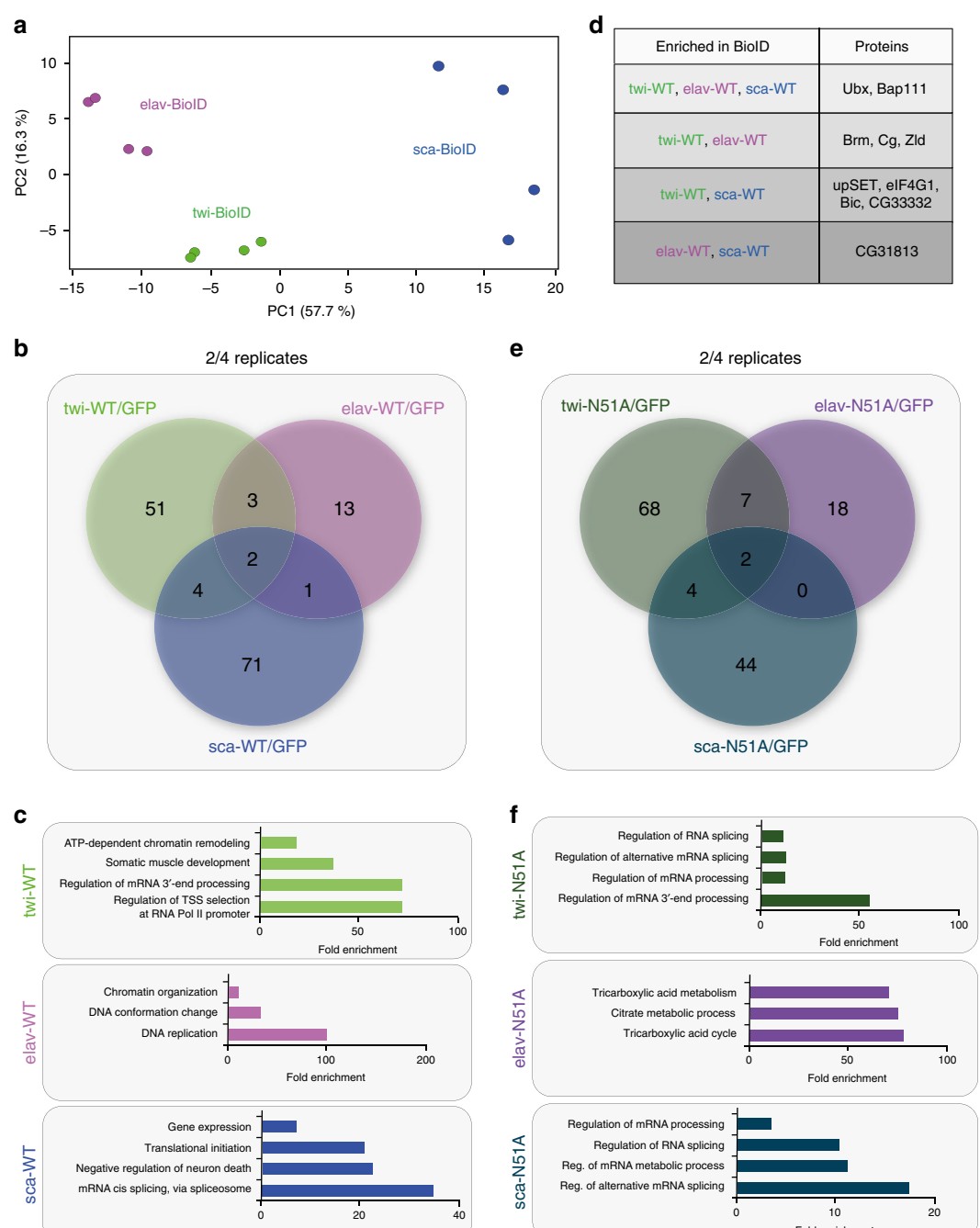

**Fig. 2 Comparison of tissue-specific Ubx BioID-interactomes. a** Principle component analysis (PCA) applied to all BioID Ubx$^{WT}$ samples identifies three clusters corresponding to mesodermal (twi-BioID), neural (elav-BioID) and neuroectodermal (sca-BioID) Ubx BioID-interactomes. This analysis also shows that the sca-BioID samples are more heterogeneous than the twi- and elav-BioID samples. **b** Venn diagram representing the overlap of Ubx$^{WT}$ interaction partners in the mesoderm (twi, green), neural (elav, purple) or neuroectodermal (sca, blue) tissue lineages. Proteins significantly enriched are present at least in 2 out of 4 replicates following calculation of WT/GFP control LFQ-log2 ratio. **c** Fold enrichment of gene ontology terms of proteins interacting with Ubx$^{WT}$ specifically in only one of the three tissue lineages (twi: 51, elav: 13, sca: 71) ($p$-value < 0.05). **d** Close-proximity partners of Ubx$^{WT}$ enriched in several tissues are shown. **e** Venn diagram representing the overlaps of Ubx$^{N51A}$ interaction partners in the mesoderm (twi, dark green), neural (elav, dark purple) or neuroectodermal (sca, dark blue) tissues (N51A/GFP control ratio). **f** Fold enrichment of gene ontology terms of proteins interacting with Ubx$^{N51A}$ specifically in only one of the three tissue lineages (twi: 68, elav: 18, sca: 44) ($p$-value < 0.05, for sca-N51A minimal raw $p$-value = 0.000233 was used). GO term $p$-value are calculated with Fisher test and FDR correction. See also Supplementary Fig. 4 and Supplementary Data 1–35. Source data are provided as a Source Data file.

interact with these proteins in a highly specific manner in the different cellular contexts.

As TF-TF pairs are central to achieve gene expression specificity[6,42–44], we next asked whether TFs were the predominant class of proteins interacting with Ubx in the different tissue

lineages. By clustering Ubx interactors based on their molecular function, we found that only a minor fraction encoded TFs (16% for mesodermal BioID-interactome), while the majority represented proteins controlling gene expression at other regulatory layers. Indeed, many of the lineage-specific Ubx interactors are

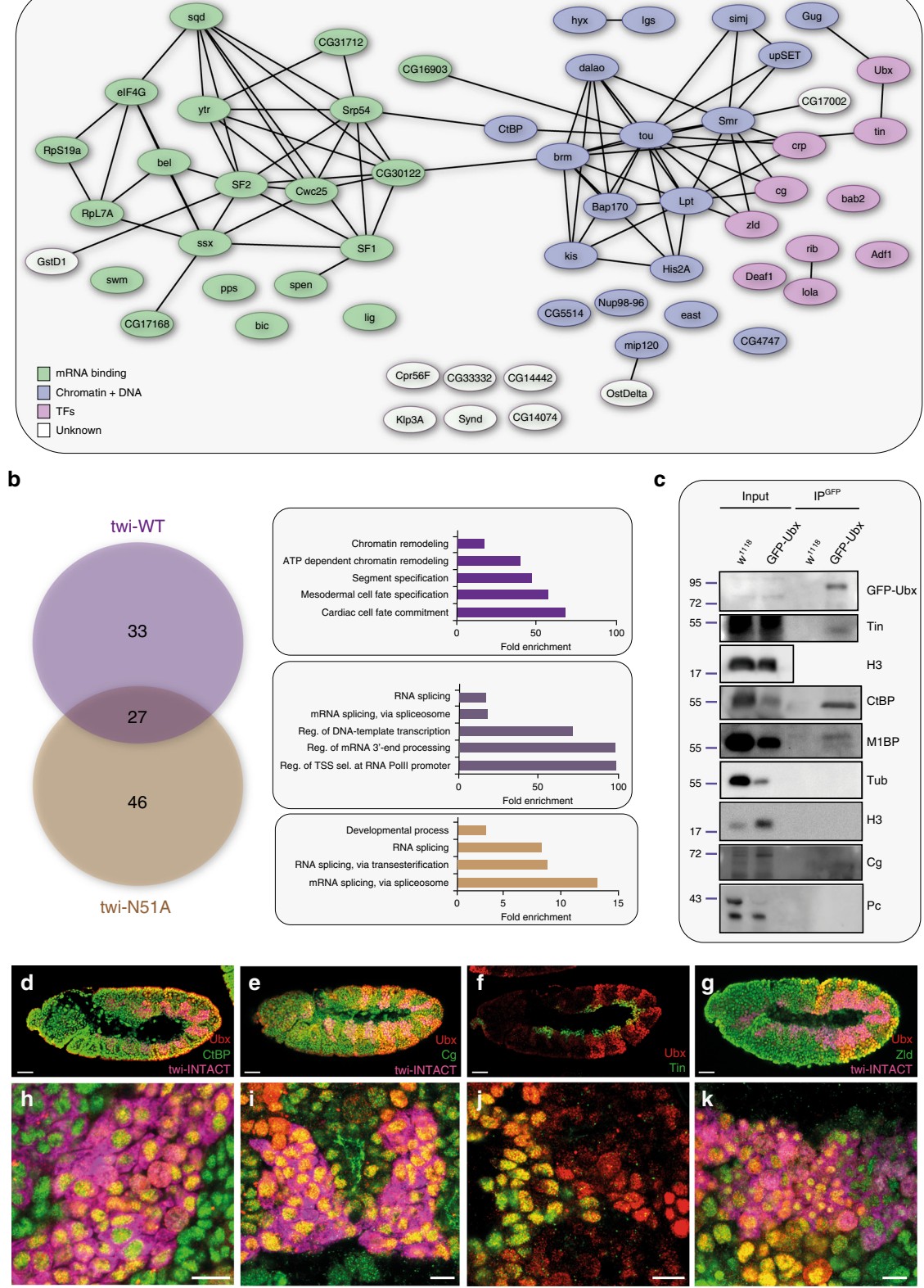

known to control co- or post-transcriptional events like RNA processing and translation (33% for mesodermal BioID-interactome) or processes that prepare the chromatin landscape for transcription, in particular chromatin remodelling events (32% for mesodermal BioID-interactome) (Figs. 2c, 3a). Consistently, STRING-based network reconstruction performed using the mesodermal Ubx close-proximity partners as input uncovered two major inter-connected grids, one related to mRNA regulation and ribonucleoprotein functions and the other one related to chromatin regulation, which included the few TFs identified as Ubx interactors (Fig. 3a, Supplementary Fig. 5).

In order to tackle how these different functions are integrated by Ubx in the nuclear environment, we analysed the compartment in which Ubx preferred to interact with its partners. To this

**Fig. 3 In vivo analysis of the mesodermal Ubx BioID-interactome. a** STRING-based reconstruction of the interaction network of all proteins identified as close-proximity partners of Ubx$^{WT}$ in the mesodermal tissue by targeted BioID. Green circles represent RNA-binding/regulatory proteins, blue circles represent chromatin and DNA-binding proteins, pink circles highlight TFs and white circles label proteins with unknown functions. **b** Left panel: Venn diagram representing the overlap of proteins enriched in close proximity to the wild-type (Ubx$^{WT}$) and mutant (Ubx$^{N51A}$) versions of Ubx protein in the mesoderm, which showed that 33 proteins interacted with Ubx preferentially on the chromatin (purple), 27 in the nucleus (dark purple) and 46 in the nucleoplasm (brown). Right panel: Fold enrichment of gene ontology terms of proteins representing the different overlap classes (chromatin, nucleus, nucleoplasm). **c** Co-immunoprecipitation of Ubx close-proximity partners from nuclear extract of control ($w^{1118}$) or *GFP-Ubx* embryos, which carry a CRISPR/Cas9 engineered version of the *Ubx* gene, *GFP-Ubx*, at the endogenous locus[2]. For co-immunoprecipitation, endogenous expression levels of all proteins were used. The input fraction is present as control (lane 1–2). Tin and CtBP were co-immunoprecipitated with GFP-Ubx (IP$^{GFP}$-lane 4), which was not the case using purified extracts from $w^{1118}$ embryos (IP$^{GFP}$-lane 3). As positive control, the known Ubx interactor M1BP was used, which was immunoprecipitated with GFP-Ubx, while Tubulin (Tub), histone H3 (H3) and Polycomb (Pc) were not pulled down with Ubx (lane 4). Protein size is indicated relative to ladder position. **d–k** Immunostaining of stage 11 embryos (3–6 h AEL) for Ubx (red) and the BioID-identified mesodermal close-proximity partners (green) CtBP (**d, h**), Cg (**e, i**), Tin (**f, j**) and Zld (**g, k**). To mark mesodermal cells, stainings were performed in the *twi-INTACT* background, which uses the tissue-specific biotinylation of the nuclear membrane protein RanGAP, and allows the detection of mesodermal nuclei by streptavidin staining (magenta). Bottom panel represents high-magnification images of mesodermal nuclei. GO term *p*-value are calculated with Fisher test and FDR correction. Images are representative of all embryos analysed over 2 sets of pooled embryos from independent collections. Scale bar = 50 µm; zoom scale bar = 10 µm. See also Supplementary Figs. 5 and 6 and Supplementary Data 36–39. Source data are provided as a Source Data file.

end, we made use of our experimental set-up and identified those proteins found in close proximity to Ubx$^{N51A}$ (Fig. 1b), the version of Ubx unable to bind DNA (N51A/GFP). We overlapped the Ubx$^{WT}$ and Ubx$^{N51A}$ BioID-interactomes and defined three protein populations: proteins interacting with Ubx preferentially on the chromatin (Ubx$^{WT}$ enriched), proteins found in close proximity to Ubx in the nucleoplasm but also on the chromatin (overlap Ubx$^{WT}$ and Ubx$^{N51A}$) and proteins interacting with Ubx in the nucleoplasm only (Ubx$^{N51A}$ enriched) (Fig. 3b, Supplementary Fig. 5, Supplementary Data 36–38). Consistently, GO terms analysis revealed that proteins interacting with Ubx on the chromatin strongly controlled chromatin-related processes in particular ATP-dependent chromatin remodelling. In contrast, proteins of the nucleoplasm/chromatin fraction preferentially regulated general processes of transcription like transcription start site selection or transcriptional initiation and post-transcriptional events like mRNA 3′-end processing or splicing. Finally, proteins of the nucleoplasm fraction were almost exclusively associated with splicing-related functions. Lineage-specific GO terms were strongly over-represented only among the chromatin population (Fig. 3b, Supplementary Fig. 5).

Together, these results demonstrated that Ubx interacted with different components of protein complexes regulating general aspects of gene expression in a lineage-specific manner. Thus, it seems that Ubx controls gene expression at multiple levels, and that the regulatory events happening at enhancers and promoters represent only one of the many layers conferring specificity to Hox TFs.

**Comprehensive validation of Ubx BioID-interactomes.** Having identified lineage-specific Ubx close-proximity partners by a proteomics-based approach, we next wanted to elucidate whether these proteins interacted with Ubx in a complex. We focused our analysis on proteins identified in the mesoderm, as we have recently characterized Ubx's function in this tissue at the chromatin level[2]. We first performed co-immunoprecipitation (co-IP) of Ubx close-proximity partners in vivo. To this end, we used embryos containing endogenously GFP-tagged *Ubx* gene and studied the interaction of GFP-Ubx with BioID candidates, for which antibodies were available. This included the transcriptional co-repressor C-terminal binding protein (CtBP), Combgap (Cg), a Zn finger TF binding to Polycomb response elements, the Zn finger TF Zelda (Zld), a known pioneer factor and the mesoderm-specific TF Tinman (Tin), a master gene of cardiac development. All four proteins were precipitated in *Drosophila* embryos by

GFP-Ubx, which was also the case for the known Ubx interactor Motif 1 binding protein (M1BP) (Fig. 3c, Supplementary Fig. 6a, Supplementary Data 39). In contrast, we could not detect an interaction between Ubx-GFP and Polycomb (Pc) recovered only by the sca-BioID and Tubulin (Tub), which was not recovered by any BioID experiment (Fig. 3c). To further characterize Ubx interactions in the mesoderm, we studied expression of Ubx, CtBP, Zld, Tin and Cg using antibody stainings as well as Brahma (Brm), the ATPase subunit of the Brahma chromatin remodelling complex, by means of a GFP fusion line in stage 10–13 embryos. As all these proteins except Tin are expressed in more than one tissue, we specifically labelled the mesoderm using the twist-INTACT transgene. Animals carrying this construct have their mesodermal nuclei biotin-labelled by the co-expressed wild-type BirA. Notably, we observed a co-localization of all five proteins with Ubx in mesodermal cells. In particular, they were co-expressed in cells of the somatic and visceral mesoderm (Figs. 3d–k, Supplementary Fig. 6b). These results demonstrated that the TFs CtBP, Cg, Zld and Tin interacted with Ubx in *Drosophila* embryos, and showed that BioID is efficient in capturing transient interactions between TF pairs in vivo.

Due to the restricted availability of antibodies, only a few BioID-identified Ubx close-proximity partners could be studied in vivo. To comprehensively validate the BioID-interactomes, we thus performed co-IP experiments in cellulo. To this end, we tested 17 BioID candidates identified in the mesoderm by overexpressing HA- or V5-tagged versions of these proteins together with nlsGFP, GFP-Ubx$^{WT}$ or GFP-Ubx$^{N51A}$ in *Drosophila* S2R+ cells. This list included the four interactors, CtBP, Cg, Zld and Tin, which we had already confirmed by in vivo co-IP, as well as the basic-helix-loop-helix TF Cropped (Crp), a factor important for muscle morphogenesis, Brahma (Brm), the ATPase subunit of the Brahma chromatin remodelling complex, Bicaudal (Bic), a protein involved in mRNA and protein localization, and a group of proteins with roles in mRNA processing, Splicing factor 1 (SF1) and Splicing factor 2 (SF2), Srp54, Cwc25, the small ribonucleoprotein particle U1 subunit 70 K (snRNPU1-70K), Small ribonucleoprotein particle protein (Smb), Scaffold attachment factor B (Saf-B), Bx-42, a splicing component that acts in the Notch pathway, SRm160, a protein important for pre-mRNA splicing and 3′ end formation, and Nucampholin (Ncm). Fifteen out of the 17 proteins were pulled down by GFP-Ubx$^{WT}$ and/or GFP-Ubx$^{N51A}$ in cellulo (Supplementary Fig. 6c–e, Supplementary Data 39) and the known Ubx cofactor Exd (Supplementary Fig. 6f). Notably, CtBP, Tin, Zld and Cg, which interacted with Ubx preferentially on the chromatin in the BioID analysis, were

pulled down more efficiently by Ubx[WT] in comparison to Ubx[N51A]. In contrast, Brm and Bic were immunoprecipitated at equal levels, while the splicing-related factors SF1, Srp54, Cwc25 and SF2 were pulled down more efficiently in co-IPs over-expressing Ubx[N51A] (Supplementary Fig. 6c, Supplementary Data 39). Having confirmed Ubx interactions in the mesoderm, we also tested two Ubx close-proximity partners identified in the neural tissue, the TFs Grh and TfAP-2 (Supplementary Fig. 6g, h). Both proteins interacted with Ubx in co-IP experiments in cellulo, again stronger with the GFP-Ubx[WT] protein (Supplementary Fig. 6h, Supplementary Data 39).

In sum, these experiments validated many of the close-proximity partners identified by BioID. It also revealed that, in contrast to mRNA-processing factors, TFs and chromatin remodelling proteins preferred to interact with the DNA-binding proficient version of Ubx, independently of cellular context. Finally, these experiments underlined again the importance of the cellular environment, as the high interaction potential of Ubx was limited to only a few specific ones in the individual lineages in vivo.

**Specificity from interaction with lineage-restricted TFs.** One question arising from this study is how Ubx can interact with different sets of functionally related and ubiquitously expressed proteins in diverse lineages. One possible explanation is the interaction of Ubx with lineage-restricted factors, which could adjust the action of Ubx to the cellular environment. We had identified a few Ubx interactors that were lineage-specifically expressed, and selected two TFs, Tin and Grh, which were enriched in the chromatin fraction of the mesodermal and neural Ubx BioID-interactomes to study their role in tissue development.

We first tested whether Tin and Grh bound the same chromatin regions as Ubx in the respective tissues. To this end, we compared genome-wide binding profiles of Tin[45], a TF exclusively active in the mesoderm, and Grh[46], a TF expressed in ectodermal and neural cells, to Ubx chromatin interactions[2]. This analysis uncovered 251 regions bound by Ubx and Tin in close vicinity in the mesodermal lineage and 401 regions co-bound by Ubx and Grh in the neural lineage among a large number of distinct binding events for all three TFs (Fig. 4a, Supplementary Table 2). Regions bound by Ubx and Tin in the mesoderm and Ubx and Grh in the nervous system, which occurred preferentially at promoters (Fig. 4b), were almost exclusive (95%, Fig. 4c). Importantly, the enhancer logic of the bound regions seemed to be different as well, as the motifs of Ubx and its known cofactor Exd were highly enriched in both Ubx-Tin and Ubx-Grh regions, while the motif of the pioneer TF Zld, a partner identified in mesodermal- and neural-BioID, was enriched exclusively among the Ubx-Tin bound chromatin sites (Fig. 4d).

Lineage-specific differences at the enhancer/promoter levels were also reflected in the genes associated with the Ubx-Tin and Ubx-Grh co-bound regions, as GO terms related to mesoderm development were over-represented among the genes bound by Ubx and Tin, while GO terms of genes bound by Ubx and Grh were associated with several tissue lineages (Fig. 4a). The latter could be due to Ubx's ability to repress the expression of alternative fate genes thereby realizing lineage development[2]. Consistently, we found that 60% of the genes targeted by Ubx and Grh in the nervous system were inactive, while the majority of genes (80%) bound by Ubx and Tin in the mesoderm were expressed (Fig. 4e). GO terms specific for the respective lineage were strongly enriched only among the active but not inactive genes bound by Ubx/Tin or Ubx/Grh (Supplementary Table 2c, d), suggesting that Ubx in combination with lineage-restricted TFs induces lineage-specific gene programs. Furthermore, the

Ubx/Tin co-bound genes were more specifically related to dorsal heart vessel and cardiac cell fate commitment compared to genes bound independently by Tin and Ubx (Supplementary Table 2a, b). This suggested that the Ubx/Tin pair is involved in defining the cardiac cell fate, thereby conferring specificity to Ubx in mesoderm development. To provide further evidence that Ubx controls the expression of genes targeted in the respective tissues, we made use of our recently published resource that identified transcriptional profiles in the mesoderm when Ubx protein was tissue-specifically degraded[2]. We found the expression of 74 out of the 367 (20%) genes bound by Ubx and Tin significantly changed in the mesoderm in the absence of Ubx (Source Data file), which included the known Tin target gene *bagpipe* (*bap*)[47] and Ubx target gene *decapentaplegic* (*dpp*)[48].

We subsequently explored the functional interplay between Ubx and Tin in more detail using *dpp* as a model[48], as we identified a Tin and Ubx ChIP peak in the well-characterized visceral mesoderm-specific *dpp* enhancer[49,50], dpp674 (Fig. 5a). Notably, *dpp* RNA expression was lost in the visceral mesoderm in the absence of Ubx, which was also the case in *tin* homozygous mutants (Fig. 5b, e, f). As the visceral mesoderm is not specified in the absence of *tin*[51], we analysed *dpp* expression in heterozygous *tin* and *Ubx* double mutants. *dpp* transcript levels were significantly reduced in heterozygous double mutants (Fig. 5d–h), showing that Ubx and Tin functions are required for the regulation of *dpp* expression. As our analysis showed that the *dpp* enhancer is bound by Ubx and Tin, we assumed that Ubx and Tin function in a combinatorial manner to activate *dpp* transcription. To support this hypothesis, we performed functional assays in *Drosophila* S2R+ cells by transiently expressing Tin, Ubx and the dpp674 enhancer, which controlled luciferase expression[48]. This analysis revealed that Ubx protein alone efficiently induced reporter gene expression even at low levels, while Tin was able to do so only at high protein concentrations (Fig. 5i). Co-expression of both proteins substantially increased luciferase expression driven by the dpp674 enhancer or by an artificial enhancer consisting of adjacent Ubx and Tin binding sites (Fig. 5i, Supplementary Fig. 7a). This effect was dependent on the homeodomains of Ubx and Tin, as reporter gene activation was not increased by Ubx and Tin protein versions unable to bind DNA (Ubx[N51A] or Tin[N51A]) (Fig. 5i, Supplementary Fig. 7b). In line, EMSA experiments confirmed the interaction of Ubx and Tin with the *dpp* enhancer, both independently and in a complex (Supplementary Fig. 7e). In sum, these results showed that Ubx and Tin functionally interacted on the *dpp* enhancer to activate gene expression.

Ubx interacts with its known cofactor Exd via two protein motifs, the hexapeptide (HX) and the UbdA domain to regulate target genes[24–26]. Thus, we asked whether one of these domains was also required for the Ubx-Tin interaction. GST pull-down experiments using purified full-length Ubx and Tin proteins revealed that Ubx directly interacted with Tin, even stronger than with Exd (Fig. 5n). To elucidate the requirements for this interaction, we generated truncated versions of Ubx (Fig. 5j). We found that only the full-length Ubx protein was highly efficient in pulling down Tin, the individual domains pulled down Tin only to a lesser extent (Fig. 5k, l), which suggested that a combination of domains are required for robust and functional interaction between Ubx and Tin. In contrast, the HX motif realized to a large extent the interaction of Ubx and Exd (Fig. 5m, Supplementary Fig. 7d), as previously described[52]. Notably, the interaction between Ubx and Tin was not influenced by the N51A amino acid exchange in the Ubx homeodomain (Fig. 5n, Supplementary Fig. 7c). These results showed that the interaction of Ubx with Tin, as with Exd, can occur independently of DNA binding. In contrast, the ability to bind DNA was required for

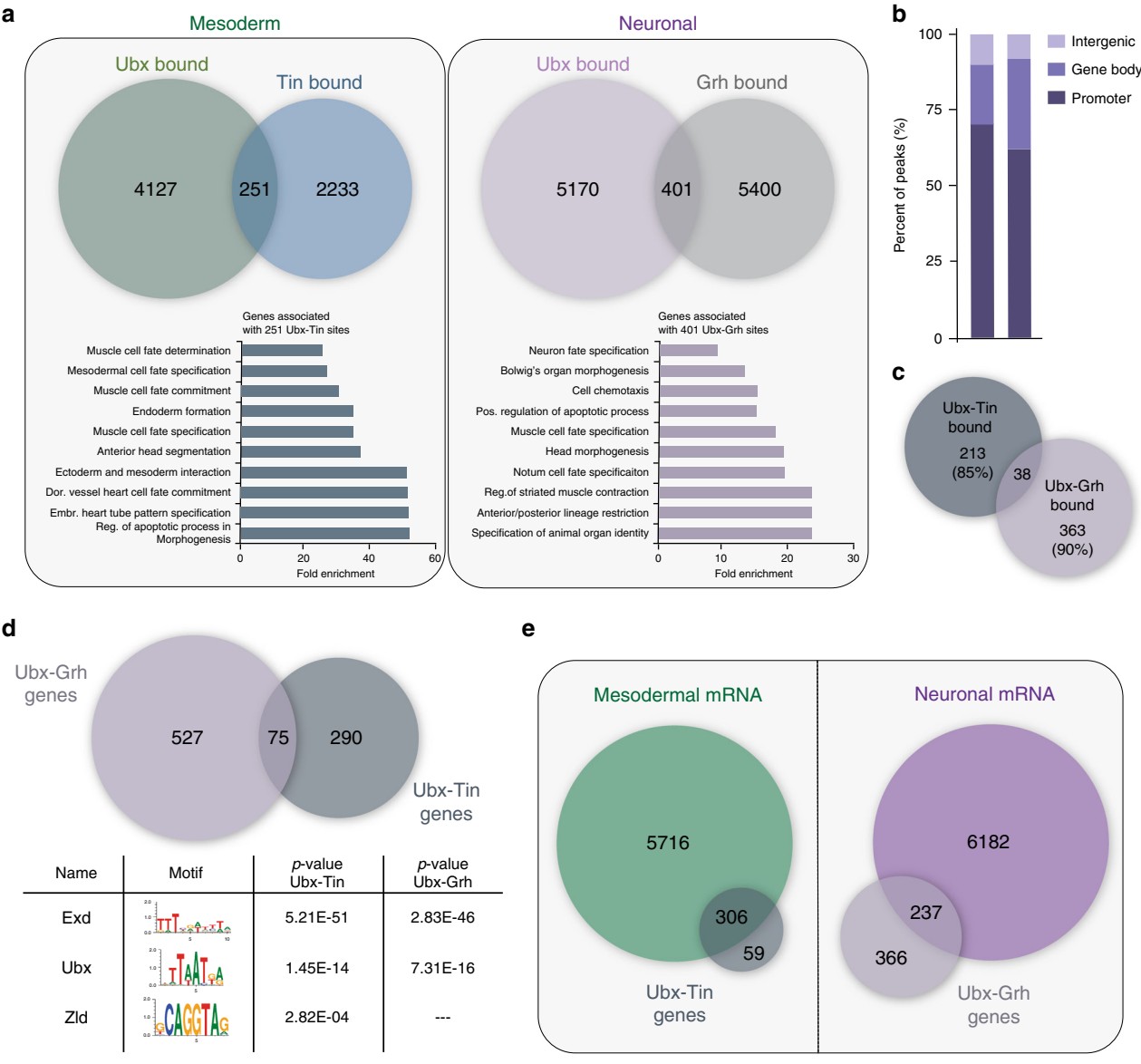

**Fig. 4 Interplay between Ubx and the lineage-restricted TFs Tin and Grh. a** Left panel: Venn diagram representing the overlap of genomic regions (1 kb) bound by Ubx (mesoderm ChIP-seq[2]) and Tin (ChIP-on-ChIP[45]). 251 regions are co-bound by Ubx and Tin in the mesoderm, which are located in the vicinity of 367 genes. Fold enrichment of gene ontology terms of genes commonly bound by Ubx and Tin in the mesoderm. Right panel: Venn diagram representing the overlap of genomic regions (1 kb) bound by Ubx (neural ChIP-seq[2]) and Grh (ChIP-seq[46]). 401 regions are co-bound by Ubx and Grh in the nervous system, which are located in the vicinity 604 genes. Fold enrichment of gene ontology terms of genes commonly bound by Ubx and Grh in the nervous system. **b** Graphical representation in percentage of the genomic localisation of region bound in common by Ubx-Tin and Ubx-Grh classified into promoter regions (−2 kb from TSS), gene bodies (5′-UTR, exon, intron, 3′UTR), and intergenic regions. **c** Venn diagram representing the overlap of genomic regions (1 kb), which are bound by Ubx-Tin and Ubx-Grh. Only 38 regions are bound by both TF combinations revealing a distinct signature of Ubx-Tin (85% exclusive) and Ubx-Grh (90% exclusive) bound *cis*-regulatory elements. **d** Top panel: Venn diagram representing the overlap of genes bound by Ubx and Tin in the mesoderm and by Ubx and Grh in the nervous system. Bottom panel: Representative motifs identified by AME motif search (MEME suite) within 1 kb region co-bound by Ubx and Tin in the mesoderm and Ubx and Grh in the nervous system. **e** Venn diagrams representing the overlap of genes bound by Ubx and Tin with the mesodermal transcriptome (mesodermal RNA-seq[2]) (left panel) and genes bound by Ubx and Grh with the neural transcriptome (neural RNA-seq[2]) (right panel). GO term and motif *p*-value are calculated with Fisher test. See also Supplementary Table 2 and Source Data file. Source data are provided as a Source Data file.

functional cooperation of both TFs in vivo (Fig. 5i, Supplementary Fig. 7b) and enhanced the interaction in cells (Supplementary Fig. 6c, Supplementary Data 39).

In sum, these results showed that Ubx cooperates with the mesodermal master regulator Tin to promote lineage development. Furthermore, our results showed that Ubx utilizes different protein domains to interact with other TFs, which we assume to be the basis of Ubx's ability to assemble cell-type-specific (co-)

transcriptional networks that function at various levels of gene expression.

**Lineage-specific functional cooperation with diverse partners.** In a final step, we sought to provide evidence that the interaction of Ubx with proteins acting at different levels of gene expression were of functional relevance and necessary for lineage

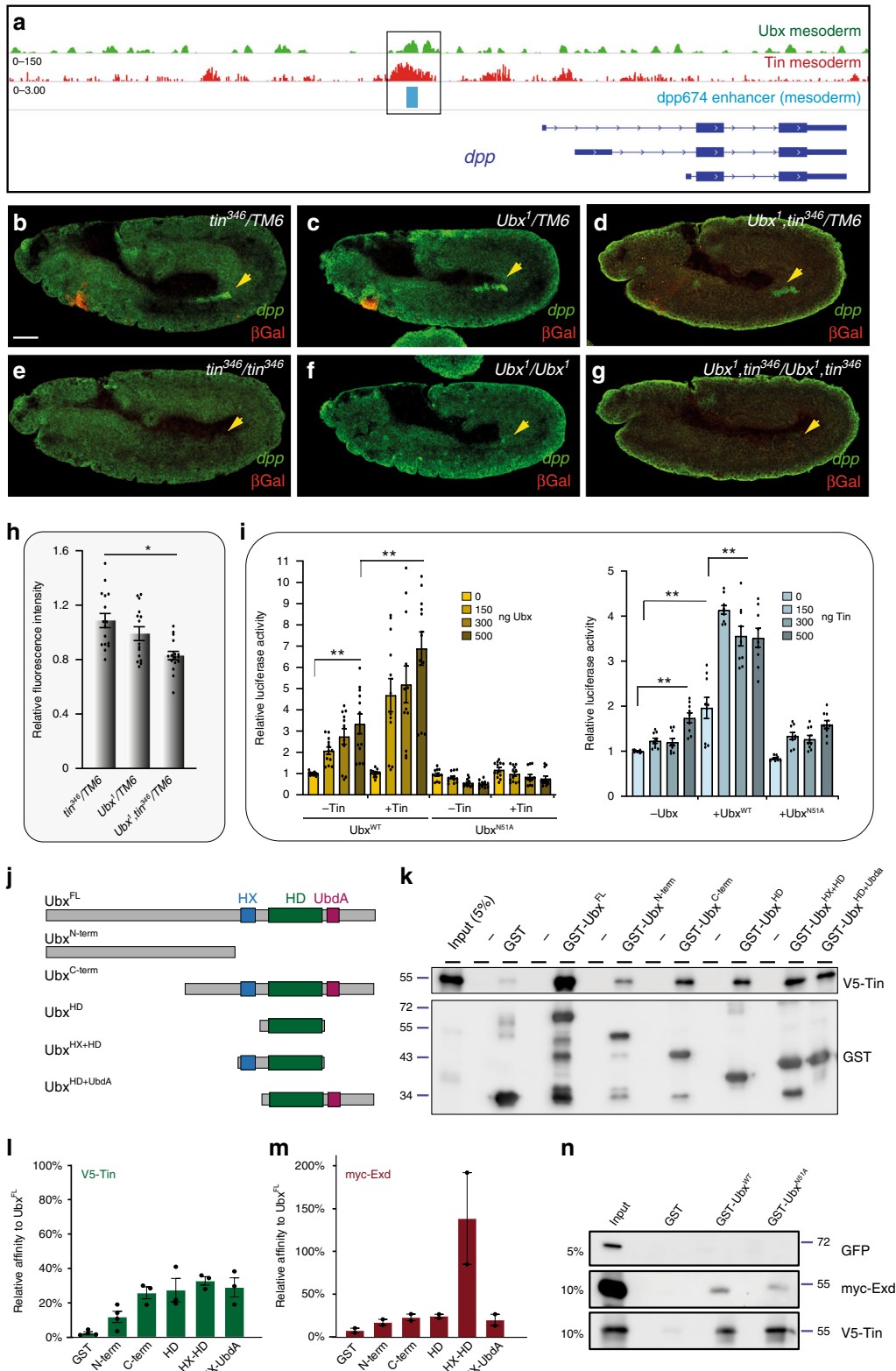

development. In addition, we wanted to test whether the specificity of interactomes identified by BioID is also reflected at the functional level. We focused our analysis on interactors identified in the mesoderm with one exception Brm, a BioID-identified interactor of Ubx in the mesoderm and nervous system (Fig. 2d). We set genetic interaction assays between *Ubx* and *tin*, *brm*, *Srp54* or *snRNPU1-70K* by crossing the *tin^346*, *brm^2*, *Srp54^DG02112* and *snRNPU1-70K^02107* null alleles into the *Ubx^1* mutant background.

Mesodermal development was studied in single as well as double heterozygous (and homozygous) stage 16 mutant embryos by characterizing the muscle morphology using Tropomyosin 1 (Tm1) (Fig. 6). We found embryos heterozygous for individual mutations to be indistinguishable from *w^1118* control embryos (Fig. 6a, c–h)[2,53], showing that a reduction of the dose of these genes did not affect the development of the mesodermal lineage. In contrast, prominent and distinct phenotypes were detected in

**Fig. 5 Direct and functional interaction between Ubx and Tin. a** ChIP-seq of Ubx[2] and ChIP-on-chip profiles of Tin[45] at the *dpp* genomic locus in mesodermal cells. Isoforms of the *dpp* gene are shown (blue) and the known *dpp* visceral enhancer (light blue). The box highlights Ubx and Tin binding to the dpp674 enhancer. **b–g** Immunostaining of stage 11 *tin³⁴⁶/TM6-Dfd>lacZ* (**b**), *Ubx¹/TM6-Dfd>lacZ* (**c**), *Ubx¹,tin³⁴⁶/TM6-Dfd>lacZ* (**d**), *tin³⁴⁶/tin³⁴⁶* (**e**), *Ubx¹/Ubx¹* (**f**) and *Ubx¹,tin³⁴⁶/tin³⁴⁶,Ubx¹* (**g**) embryos for *dpp* mRNA (green) and β-Galactosidase protein (red). Images are representative of all embryos analysed (*n* = 15) per genotype over 2 sets of pooled embryos from independent collection. **h** Quantification of relative signal intensity of *dpp* mRNA levels shows significant expression changes between *tin³⁴⁶/TM6-Dfd>lacZ* and *Ubx¹,tin³⁴⁶/TM6-Dfd>lacZ* heterozygous mutants (*n* = 15 independent embryos). **i** S2R+ cells were co-transfected with a *dpp674*-containing plasmid driving luciferase expression, myc-Ubx and V5-Tin encoding plasmids (100 ng). Increasing amounts of Ubx^WT or Ubx^N51A (left) or Tin expressing plasmids (right) was used. Transfection efficiency was normalized with Renilla activity originating from co-transfected pRT-TK plasmid. Results are indicated relative to basal activity of the *dpp674*-luciferase plasmid. Graphics represent mean +/− sem of three (*n* = 3) independent experiments performed in triplicates. **j** Schematic of Ubx fragments used for GST in vitro pull-down assays. Hexapeptide (HX) motif is highlighted in blue, the Homeodomain (HD) in green and the UbdA motif in dark-red. **k** Pull-down assay using the indicated GST-fused Ubx derivatives and in vitro purified Tin. Input is loaded as indicated. **l, m** Quantification of interactions relative to GST-Ubx^FL (full length) signal is indicated in (**l**) for V5-Tin (*n* = 3) and in (**m**) for myc-Exd (*n* = 2 independent experiments) as mean ± SD. **n** Pull-down assay using GST-fused Ubx derivatives and in vitro His-purified GFP (negative control), myc-Exd (positive control) and V5-Tin proteins. Input is loaded as indicated. Protein size is indicated relative to ladder position. Pull-down assays showed direct interaction of Ubx^WT as well as Ubx^N51A with Exd and Tin but not with GFP (lane 3–4). Ubx-Tin interaction is at least 10 times stronger than Ubx-Exd interaction as exemplified by intensity of signal in pull-downs compared with input (comparison of lane 3–4 with 1). Statistical tests were performed with one-way ANOVA (**p* < 0.01, *p* < 0.05). Scale bar = 50 μm. See also Supplementary Fig. 7. Source data are provided as a Source Data file.

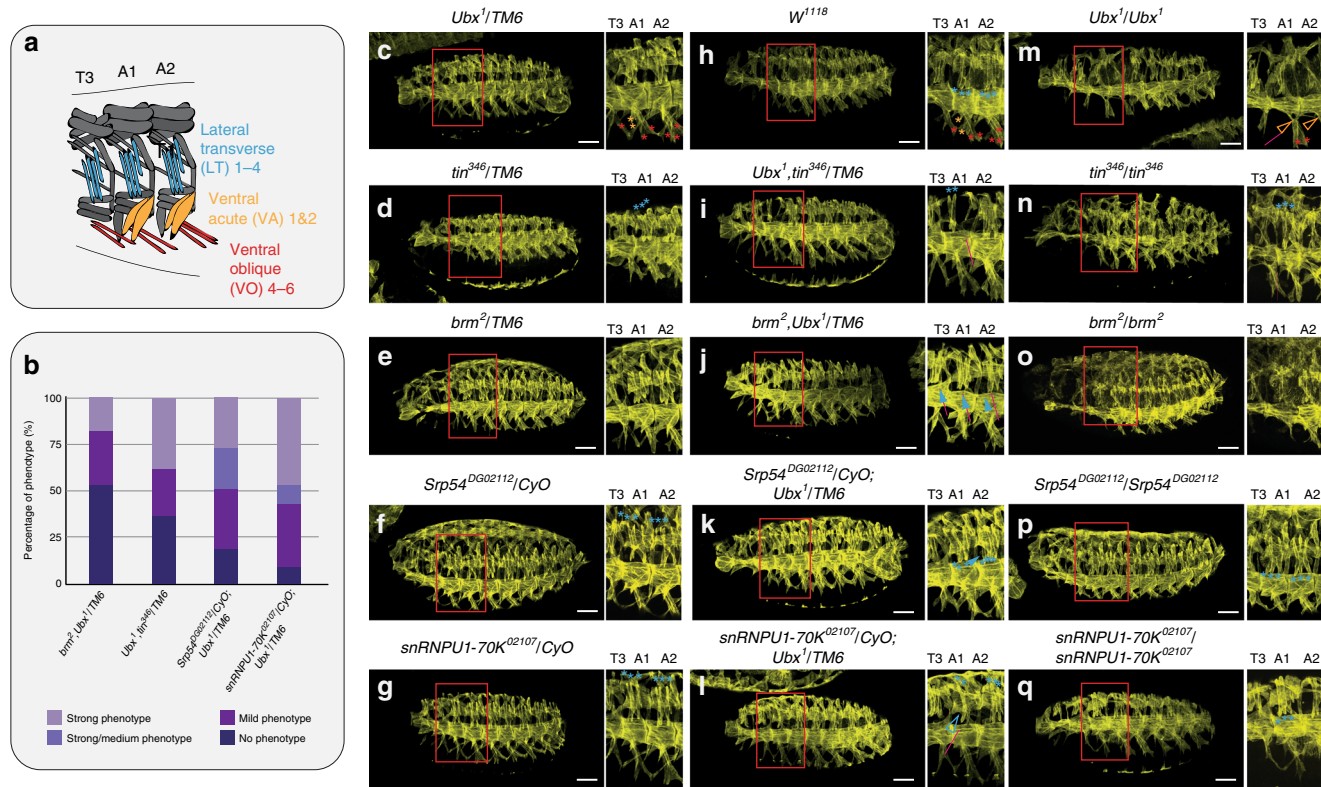

**Fig. 6 Mesoderm-specific functional cooperation of Ubx with BioID interactors. a** Schematic illustration of the muscle patterns in the thoracic segment 3 (T3) and the first two abdominal segments (A1, A2). Lateral transverse (LT) muscles are highlighted in blue, ventral acute (VA) muscles in orange and ventral oblique (VO) muscles in red, corresponding muscles in all the images are marked with asterisks or arrows in the indicated colours. **b** Quantification of muscle phenotypes in the indicated genotypes. Phenotype–genotype association was done by using LacZ staining driven by the marked balancer chromosomes. Strong and strong/medium phenotypes correspond to double homozygous mutants and mutants homozygous for one allele and heterozygous for the other. Embryos heterozygous for both mutant alleles showed either a mild phenotype or a hardly visible phenotype, categorized as "no phenotype". **c–q** Immunostainings of stage 16 embryos of the indicated genetic backgrounds with tropomyosin (Tm1) to visualise the muscle pattern. While single heterozygous mutants (**c–g**) did not display any obvious muscle phenotype, double heterozygous mutants (**h–l**) had changed muscle morphology, which was distinct from the single homozygous mutant phenotypes (**m–q**). A significant loss of the lateral transverse muscles is detectable in *Ubx¹,tin³⁴⁶/TM6-Dfd>lacZ* (**i**), *Srp54^DG02112/CyO-wg>lacZ;Ubx¹/TM6-Dfd>lacZ* (**k**) and *snRNPU1-70K^02107/CyO-wg>lacZ;Ubx¹/TM6-Dfd>lacZ* (**l**) double heterozygous mutants indicated by blue asterisks and an open arrowhead. *brm²,Ubx¹/TM6-Dfd>lacZ* (**j**) double heterozygous mutant embryos show an aberrant muscle pattern indicated by the closed blue arrowheads. **m** *Ubx¹* homozygous mutants show a homeotic transformation of A1 and A2 to T3 identity indicated by the missing asterisks (red, orange), while *Srp54^DG02112* (**p**) and *snRNPU1-70K^02107* (**q**) homozygous mutants show discrete muscle phenotype characterized by thinner muscles except *tin³⁴⁶* embryos that present a total loss of body wall muscle affecting the general aspect of embryos. Maximum Z-projections of lateral view are presented. Images are representative of 50 embryos analysed per genotype over 3 sets of pooled embryos from independent collection as quantified in **b**. Scale bar: 50 μm. See also Supplementary Figs. 8–10. Source data are provided as a Source Data file.

the muscle lineage in double heterozygous mutants. While lateral muscles but not the ventral oblique muscles (VO4-VO6) were either lost or malformed in the first two abdominal segments (A1, A2) in $Ubx^1,tin^{346}$, $brm^2,Ubx^1$ and $snRNPU1-70K^{02107};Ubx^1$ heterozygous mutants (Fig. 6a, i–j, l), an extra transversal muscle was found in $Srp54^{DG02112};Ubx^1$ double heterozygous mutants (Fig. 6k). Moreover, these phenotypes were different from those observed in embryos carrying individual homozygous null alleles. For example, $Ubx^1;Ubx^1$ mutants displayed homeotic transformation of A1 and A2 muscle pattern, including the absence of ventral oblique muscles (VO4-VO6) characteristic for thoracic segments (Fig. 6m)[54], a phenotype not found in any of the double heterozygous mutants (Fig. 6i–l). In line, homozygous $brm^2$, $snRNPU1-70K^{02107}$ as well as $Srp54^{DG02112}$ mutants had thinned transversal muscles (Fig. 6o–q), which was not the case in heterozygous combinations with the $Ubx^1$ allele (Fig. 6j–l). Importantly, we did not detect a phenotype in the neural lineage for the double heterozygous mutants of Ubx and the mesoderm-specific interactors (Tin, Srp54, snRNPU1-70K), as neither the number of neuroblasts (NBs) within the ventral nerve chord (VNC) nor the innervation of the ventral-lateral muscle 1 (VL1) of abdominal segments, both affected in Ubx mutant embryos, were changed in comparison to control animals (Supplementary Figs. 8–10). In contrast, double heterozygous $brm^2,Ubx^1$ mutant embryos were characterized by additional neuroblasts in the A1 segment (Supplementary Figs. 9 and 10), which is consistent with our data on Ubx interacting with Brm in the mesodermal and neural lineages. Finally, we also studied the interaction between Ubx and its BioID-identified neural partner Grh, as the two proteins co-localised in vivo and interacted by co-IP in cells (Supplementary Fig. 6g, h). Using Dpn stainings as read-out[40,55–57], we found that single $Ubx^1$ and $grh^{IM}$ homozygous mutants displayed supernumerary NBs in the A1 segment (Ubx, +4NBs) and in all abdominal segments (Grh), while single heterozygous mutants did not show a significant change in NBs number (Supplementary Fig. 9). In contrast, $grh^{IM};Ubx^1$ double heterozygous mutant embryos exhibited additional NBs in A1 and A2 segments, revealing a functional cooperation between Ubx and Grh during programmed cell death[56,58,59]. This interaction is of functional importance only in the neural lineage, as the muscle pattern was unaltered in $grh^{IM};Ubx^1$ double heterozygous mutant embryos (Supplementary Fig. 9h–j).

In sum, these results demonstrated that the Hox TF Ubx functions not only via the interaction with other TFs at *cis*-regulatory modules but uses a whole battery of proteins acting at different levels of gene expression. Importantly, most of the interactors are commonly expressed, nonetheless the interactions with Ubx and the functional outputs are highly lineage- and factor-specific, enabling Ubx to control different aspects of development in a precise manner in diverse lineages.

## Discussion

Proteins interact with a multitude of partners in a highly specific yet dynamic and context-dependent manner, which is detrimental for a cell to adopt and maintain its appropriate fate. So far it has been challenging to capture these diverse and transient interactions due to the lack of sensitive-enough methods, which unbiasedly identify factors in close proximity in different cellular contexts in the living organism. To fill this gap, we have designed a targeted proximity proteomics approach by combining BioID[34] and the GAL4-UAS system[38]. We selected the Hox TF Ubx and the mesodermal, neural and neuroectodermal lineages as a model to verify the functionality of the system. Using this approach, which requires protein overexpression, we identified Ubx interactomes specific to each lineage. By comparing the Ubx

interactomes identified by BioID to proteins known to physically interact with Ubx[60], we found only a small overlap (Supplementary Fig. 11). This is in line with recent studies showing that different methods capture variable types of protein–protein interactions, which are all biologically relevant[32]. Our data support this notion, especially as we have validated a substantial number of Ubx interactions by co-IP. Analysing the proteins identified by other methods in more detail revealed that they were enriched for chromatin interacting proteins, in particular TFs. This bias is, however, not a result of Ubx's preference to interact primarily with other TFs but intrinsic to the dataset, as it is largely based on Bi-molecular fluorescence complementation (BiFC) screens, which used pre-selected TFs to test their ability to interact with Ubx and other Hox proteins[52,61]. Thus, targeted BioID is a valuable and powerful method and ideally complements other approaches, as it captures dynamic, weak and specific interactions in vivo in an unbiased manner.

Our study, which analysed independent tissue lineages of comparable developmental stages, revealed that Ubx interacted with a largely non-overlapping set of proteins in the different cellular contexts. In contrast, Ubx interacted equally well with all the proteins identified by BioID in cellulo. These results demonstrated first, that the Hox protein Ubx has an intrinsically high interaction potential, which has been noted before[52,61,62]. Secondly, this high interaction potential is restricted to a few specific ones in vivo, where the cellular context dictates the type of interactions. Importantly, our genetic interaction studies demonstrated that these context-specific interactions are of functional importance in vivo and indeed active only in specific lineages. One question arising from this behaviour is how interaction specificity, which allows a precise matching of Hox function and activity to the cell type and developmental stage, is achieved. It is known that Ubx protein, like many other TFs, harbours intrinsically disordered domains that are important for selecting interacting partners[63–65]. Thus, the few lineage-restricted Ubx interactors identified in this study, in particular Tin or Grh, could be responsible for Ubx's differential interaction potential by binding to these intrinsically disordered domains. They could enforce lineage-specific protein conformations that can only be bound by a subset of the many Ubx interactors. In line, we found that the interaction of Ubx with Tin required the full-length Ubx protein, and was not driven by previously characterized structured domains like the homeodomain or HX motif. In addition, it is known that intrinsically disordered domains are the predominant sites of post-translational modifications[66]. They can have a pronounced effect on the structural and physico-chemical properties of a protein, modulating the composition of protein complexes. Thus, it is tempting to speculate that the different interactomes assembled by Ubx in the mesodermal and neural lineages are also dependent on specific post-translational modifications, which are cell type- and stage-specifically written on intrinsically disordered domains of Ubx. Consistently, it is now more and more realized that Hox TFs are heavily modified at the post-translational level[67,68]. In future, it will be crucial to characterize Ubx-Tin and Ubx-Grh complexes on the structural-functional level and to study cell type-specific post-translational modifications of Ubx in vivo to resolve the specificity problem intrinsic to Hox TFs.

Another striking finding of our study is that although Ubx interactions were distinct in the different tissue types, most of the proteins were not lineage-specifically expressed but active in many cell types. Indeed, the majority of Ubx interactors encoded ubiquitously expressed proteins, which are part of complexes controlling general aspects of gene expression. This included regulators of the chromatin landscape with an emphasis on chromatin remodelling components, proteins of the Polycomb

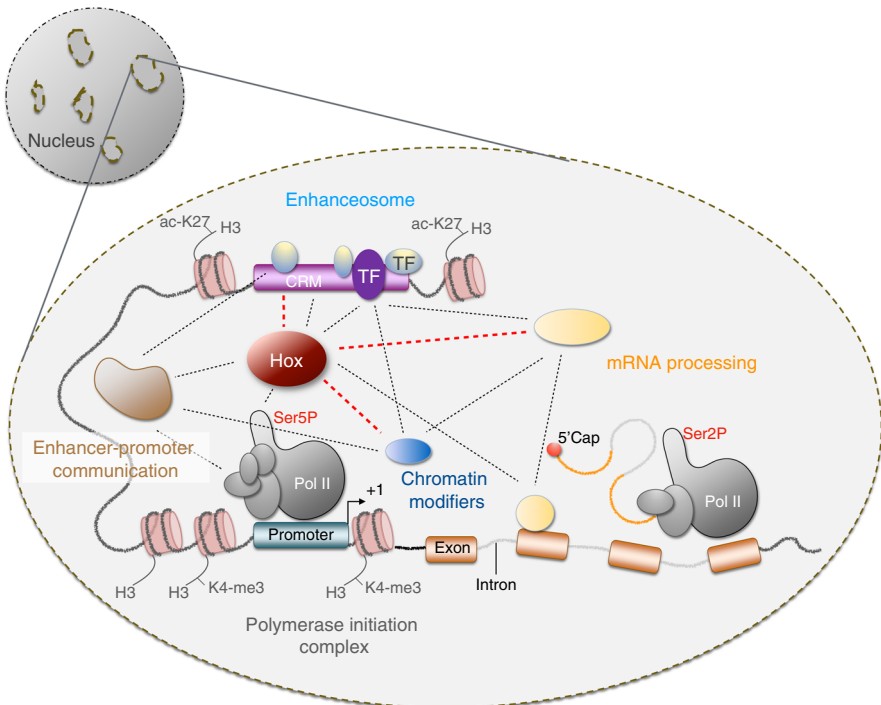

**Fig. 7 Model of tissue-specific Ubx transcriptional interactive networks.** Ubx specificity and functional diversity could be the result of tissue-specific transcriptional networks assembled at the cell-specific level but also at several layers of gene expression. From *cis*-regulatory module (CRMs) to mRNA processing, it may act as a transcriptional platform integrating multi-layered and inter-connected interactions (dot-lines) with multi-lineage (shadow-grey) or cell-restricted (purple) TFs, regulators of chromatin-contact (shadow brown), basal RNA-polymerase II machinery (grey, pol II), chromatin modifiers (blue) and mRNA-processing regulators (orange). Interactions are mainly formed with ubiquitously expressed proteins in a tissue-specific manner (red line) and a few lineage-restricted TFs (purple TF proteins) that might promote specific conformational changes. In the context of transcriptional micro-environments, this might be the first layer of cell-specificity driving the assembly of higher-order interactive networks at multiple transcriptional levels in a precise spatial and temporal context. Image adapted from Carnesecchi et al.[12] with permission.

complex, and major regulators of mRNA processing and protein translation. It is well-described that mRNA processing is a co-transcriptional process[69,70]. Moreover, the chromatin environment affects transcription at different levels by modulating enhancer accessibility[71] or the speed rate of the RNA-polymerase II through gene bodies[72]. Similarly, recent studies revealed that chromatin regulators, such as components of the remodelling complex SWI/SNF, interact with snRNP proteins[73]. Our proteomics and functional data now showed that all these proteins, which act at different control levels of gene expression programs, converge on the Hox TF Ubx (Fig. 7). Thus, Ubx seems to act as a protein platform that integrates in a highly flexible manner multiple regulatory inputs, possibly via its intrinsically disordered domains, to realize the many different yet specific outputs. Consistent with that view, it has been shown recently that Ubx forms dynamic sub-nuclear protein clusters, so-called micro-environments, that promote gene expression in vivo[74]. In that respect, the currently discussed phase separation model for transcriptional regulation is of particular interest[75–77]. It represents a concentration of regulatory proteins in active nuclear sub-domains driven by weak and dynamic interactions, in defined cellular condensates that we seem to have captured in vivo (Fig. 7). In the future, it will be highly relevant to relate the dynamic Ubx transcriptional hubs with the lineage-specific interaction networks identified in this study to elucidate how such multivalent interactions control precise gene expression programs, which realize and maintain specific cell fates.

## Methods

**Fly line and materials**. For the BioID, nlsGFP, Ubx^{WT} and Ubx^{N51A} (site directed mutagenesis) were generated, cloned in pUAST-attB-myc-BirA*-GGSGG- (BioID

cloned from #35700, Addgene) and the constructs were integrated stably on the third chromosome using the Bestgene service. The subsequent *UAS-BioID* lines were crossed in the *twist-GAL4 (twi)*, *elav-GAL4*, *scabrous-GAL4 (sca)* background to generate *elav-GAL4;;UAS-mB*nlsGFP*, *sca-* and *twi-GAL4;UAS-mB*nlsGFP* stable lines. For *UAS-mB*Ubx^{WT}* and *Ubx^{N51A}*, males were crossed with female containing driver-GAL4. Plasmids generated for the study, oligonucleotides and fly lines (generated, generously provided or from Bloomington center) are listed, referenced in Supplementary Table 3 and available upon request. myc-BioID2-MCS was a gift from Kyle Roux (Addgene plasmid # 74223; http://n2t.net/addgene:74223; RRID:Addgene_74223). pcDNA3.1 mycBioID was a gift from Kyle Roux (Addgene plasmid # 35700; http://n2t.net/addgene:35700; RRID: Addgene_35700).

**Cell culture and transfection**. S2R+ Drosophila cells (generously provided by the Tobias Dick lab (DKFZ Heidelberg), originated from Drosophila Genomics Resource Center) were maintained at 25 °C in Schneider medium supplemented with 10% FCS, 10 U/ml penicillin and 10 μg/ml streptomycin. Cells were simultaneously seeded and transfected with Effectene (Qiagen) according to the manufacturer's protocol. Cells were harvested in Phosphate Buffered Saline (PBS) and pellets were resuspended in RIPA buffer supplemented with protease inhibitor cocktail (Sigma-Aldrich). For interaction assay, $10 \times 10^6$ cells were seeded in 100 mm dishes. Biotin treatment (Sigma) was applied for 24 h after transfection. Cells were harvested in Phosphate Buffered Saline (PBS) after 48 h of transfection and pellets were resuspended with lysis buffer supplemented with protease inhibitor cocktail (Sigma-Aldrich) and 1 mM of DTT. For luciferase assays, cells were co-transfected with pRT-TK-Renilla or pActin-β-Galactosidase plasmid (Promega) for normalization. Cells were harvested 48 h after transfection and luciferase assay for Beta-galactosidase, Renilla and Firefly were analysed using beta-Galactosidase or Dual-luciferase detection kit (Promega). Plasmids are listed in Supplementary Table 3a.

**Co-immunoprecipitation of cell and embryos lysate**. For co-immunoprecipitation assays, cells were harvested in Phosphate Buffered Saline (PBS) and pellets were resuspended in NP40 buffer (20 mM Tris pH 7.5, 150 mM NaCl, 2 mM EDTA, 1% NP40) and treated with Benzonase (Sigma). GFP-Trap beads (Chromotek) were added to the protein extract, incubated for 2 hours and washed five times with NP40 buffer. For in vivo interaction, overnight collection of embryos was dechorionated,

fixed with 3.2% formaldehyde and collected in PBS supplemented with Tween 0.1%. Pellets were resuspended in buffer A (10 mM Hepes pH 7.9, 10 mM KCl, 1.5 mM MgCl₂, 0.34 M sucrose, 10% glycerol) and dounced 25–30 times with loose- and 5 times with tight-pestle. Lysates were incubated with 0.1% Triton and centrifuged. Nuclear pellet were then resuspended with buffer B (3 mM EDTA pH 8, 0.2 mM EGTA pH 8), sonicated (Picoruptor, Diagenode), and treated with Benzonase. Four to five milligrams of nuclear lysates were diluted in NP40 buffer (20 mM Tris pH 7.5, 150 mM NaCl, 2 mM EDTA, 1% NP40) and incubated overnight with 40 µl of GFP-Trap beads. Beads were then washed five times with NP40 buffer and all samples were resuspended in Laemmli buffer for immunoblotting analysis. All buffers were supplemented with protease inhibitor cocktail (Sigma), 1 mM of DTT and 0.1 mM PMSF. Input fractions represent 1–10% of the immunoprecipitated fraction.

**SDS-page and immunoblotting.** For western blot analysis, proteins were resolved on 8–15% SDS-PAGE, blotted onto PVDF membrane (Biorad) and probed with specific antibodies after saturation. The antibodies (and their dilution) used in this study were Ubx (home-made, 1/200), Cg (generously provided by William Brook, 1/500), Histone 3 (1791 Abcam, 1/10,000), GFP (A11122 Life Technologies, 1/3000), myc (SC40 Santa Cruz, 1/500e), Streptavidin-HRP (RPN1231 GE-healthcare, 1/500e), CtBP (generously provided by David Arnosti, 1/500e), Zld (generously provided by Julia Zeitlinger, 1/500e), Tin (generously provided by Manfred Frasch, 1/1000e), M1BP (generously provided by Andy Saurin, 1/500e), Pc (generously provided by Jürg Müller, 1/200e), Tubulin (MCA77G Serotec/Biorad, 1/2000e), HA (3724 Cell Signaling, 1/3000e), V5 (13202 Cell Signaling, 1/3000e), GST (2624 Cell Signaling, 1/5000e), Flag-M2 (F1804 Sigma, 1/1000e), Med19 (generously provided by Muriel Boube, 1/500e). Developing was performed using chemiluminescence reaction (ECL, GE-Healthcare) with secondary coupled to HRP (Promega, 1/5000e).

**Protein purification and GST pull-down.** All His-tagged and GST-tagged proteins were cloned for this study in pET or pGEX-6P plasmids, respectively. His- and GST-tagged proteins were produced from BL-21 (RIPL) bacterial strain, purified on Ni-NTA agarose beads (Qiagen) or Gluthatione-Sepharose beads (GE-Healthcare) and quantified by Coomassie staining. His-tagged proteins were specifically eluted from the beads with Imidazole. In vitro interaction assays were performed with equal amounts of GST or GST fusion proteins in affinity buffer (20 mM HEPES, 10 µM ZnCl₂, 0.1% Triton, 2 mM EDTA) supplemented with NaCl, 1 mM of DTT, 0.1 mM PMSF and protease inhibitor cocktail (Sigma). Proteins produced in vitro were subjected to interaction assays for 2 h at 4 °C under mild rotation. Bound proteins were washed four times and resuspended in Laemmli buffer for western blot analysis. Input fraction was loaded as indicated.

**EMSA.** The 5′-Cy5-labelled complementary oligonucleotides (Eurofin) commercially produced were annealed before reaction. The sequences used for this study were the following: Ubx sites: Cy5-5′-TTCAGAGCGAATGATTTATGACCGG TCAAG-3′. For dpp-labelled probes, PCR-labelling has been used for generating DNA fragments of the 675 bp enhancer with the following primers: F1 (188 bp) Cy5-5′-GGATCCGAAATAGTTAGTGTA-3′ and Cy5-5′-ACCAGGGGTTCTTC TTCGAC-3′, F2 (192 bp) Cy5-5′-CCTGAATCCCGACACAACCC-3′ and Cy5-5′-TAAAACAACGGATCGTGCAT-3′, F3 (150 bp) Cy5-5′-CAATCGCTGTAAAT AAATAG-3′ and Cy5-5′-CGGCAAATTGCAGCGCGCAT-3′, F4 (145 bp) Cy5-5′-CCATTCGGCTCAACAGTTAT-3′ and Cy5-5′-GTGGGCCACAAATCAA ATTG-3′. The F3-fragment was further used for the study. The binding reaction was performed for 20 min in a volume of 30 µl containing 1x Binding Buffer (20 mM Hepes pH 7.9, 1.4 mM MgCl₂, 1 mM ZnSO₄, 40 mM KCl, 0.1 mM EDTA, 5% Glycerol), 0.2 µg Poly(dI-dC), 0.1 µg BSA, 10 mM DTT and 0.1% NP40. For each reaction His-purified proteins were used. Antibodies were added as indicated for 10 additional min (13202, Cell Signaling, V5; 2396, Cell Signaling, MBP). Separation was carried out (200 V, 50 min for 30 bp, 150 V, 1h15 for >100 bp probes) at 4 °C on a 6% acrylamide gel in 0.5x Tris-borate-EDTA buffer to visualize complex formation by retardation. Cy5-labelled DNA-protein complexes were detected by fluorescence using INTAS Imager.

**BioID.** Similar to co-immunoprecipitation, dechorionated embryos (staged at 29 °C, according to Fig. 1b) were rinsed with Embryo Collection Buffer (0.7% NaCl; 0.1% Triton) and embryos pellet were frozen (−80 °C). Pellets were resuspended in buffer A, dounced 40 times with loose-, 10 times with tight-pestle and transfer through miracloth membrane to new tube. Lysates were incubated with 0.1% Triton and centrifuged 1500 × g, 5 min at 4 °C. Nuclear pellets were washed with Buffer A and centrifuged once more. Nuclear pellets were then resuspended with buffer B, sonicated (Picoruptor, Diagenode), treated with Benzonase and centrifuged at maximum speed. For affinity purification (AP), 3–6 mg of nuclear extracts were used and two AP were combined for each samples. Protease-resistant streptavidin beads (patent pending) were equilibrated with two PBS washes and resuspended in RIPA buffer supplemented with 1% SDS (50 mM Tris pH 8, 150 mM NaCl, 0.5% sodium deoxycholate, 1% NP40, 1% SDS). Clear nuclear extracts were incubated with 60 µl of streptavidin beads in a final volume of 1.5 ml RIPA-SDS for 4–5 h. Beads were then washed twice with SDS-Buffer (10 mM Tris.

HCl, 1 mM EDTA, 1% SDS, 200 mM NaCl), twice with RIPA-SDS and twice with acetonitrile buffer (20% acetonitrile in MS-grade water).

**Mass spectrometry preparation.** Streptavidin beads were resuspended in 14 µl of ammonium bicarbonate 50 mM and proteins were subjected to reduction with 1 µl DTT (100 µM) at 60 °C for 15 min followed by alkylation with 1 µl of Iodoacetamide (IAA 200 mM) for 45 min at room temperature in the dark. Protein digestion was performed on beads with a Trypsin/LysC mix (Promega, V5071) at 37 °C for 14 h. Peptides were de-salted using the SP3 protocol as previously described[78–80]. Peptides were eluted in trifluoroacetic acid (TFA) 0.1% and loaded on a trap column (Thermo acclaim pepmap 100, 100 µm × 20 mm) (PepMap100 C18 Nano-Trap 100 µm × 20 mm) and separated over a 50 cm analytical column (Waters nanoEase BEH, 75 µm × 250 mm, C18, 1.7 µm, 130 Å) using the Thermo Easy nLC 1200 nanospray source (Thermo EasynLC 1200, Thermo Fisher Scientific). Solvent A was water with 0.1% formic acid and solvent B was 80% acetonitrile, 0.1% formic acid. During the elution step, the percentage of solvent B increased in a linear fashion from 3 to 8% in 4 min, then increased to 10% in 2 min, to 32% in 68 min, to 50% in 12 min and finally to 100% in a further 1 min and went down to 3% for the last 11 min. Peptides were analyzed on a Tri-Hybrid Orbitrap Fusion mass spectrometer (Thermo Fisher Scientific) operated in positive (+2 kV) data dependent acquisition mode with HCD fragmentation. The MS1 and MS2 scans were acquired in the Orbitrap and ion trap, respectively, with a total cycle time of 3 s. MS1 detection occurred at 120,000 resolution, AGC target 1E6, maximal injection time 50 ms and a scan range of 375–1500 m/z. Peptides with charge states 2–4 were selected for fragmentation with an exclusion duration of 40 s. MS2 occurred with CE 33%, detection in topN mode and scan rate was set to Rapid. AGC target was 1E4 and maximal injection time allowed of 50 ms. Data were recorded in centroid mode.

**Cuticle preparation.** Short egg collection (3–8 h at 29 °C) followed by 12–18 h of additional development was dechorionated and transferred in glass vial with 4 ml Heptane/4 ml Methanol and shaked vigorously. Embryos/larvae were then washed four times with methanol and four times with water containing 0.1% tween. Larvae were subsequently mounted between glass in Hoyer's medium and incubated for 2–3 days at 60 °C. Photographs were performed with Axio Imager.M1 (Zeiss), objective ×40 using brightfield. For microscopy, all analysis were performed with Fiji (Fiji is Just ImageJ).

**Immunofluorescence, in situ hybridization and imaging.** For immunostainings[2], embryos were dechorionated, fixed with formaldehyde supplemented with heptane and vitelline membrane removed using methanol. Embryos were washed in PBS-tween 0.1%, blocking was performed with BSA 1% in PBS-tween and primary antibodies were incubated overnight. Secondary antibodies coupled to fluorescent protein (1/200e, Jackson) were further incubated for 2 h the following day and embryos mounted in Vectashield-DAPI. The following antibodies were used: Elav (1/50, DSHB), GFP (1:300, Invitrogen, A11122), Myc (1/300, Santa Cruz, SC40), Ubx (1/100e, Home-made), Cg (1/200e, generously provided by William Brook), CtBP (1/1000, generously provided by David Arnosti), Zld (1/500e, generous gift from Julia Zeitlinger), Tin (1/1000e, generous gift from Manfred Frasch), Grh (1/100e, generous gift from Bill McGinnis), Beta-Galactosidase (1/1000e, Promega, Z3783), Digoxigenin (1/1000e, Roche), Tm1 (1/1000e, Abcam, ab50567), Fasciclin2 (1/50e, DSHB), Engrailed (En) (1/2.5, DSHB), Deadpan (Dpn, generous gift from Jürgen Knoblich and Ana Rogulja-Ortmann). Streptavidin (1/500e, Perkin-Elmer) was revealed with the TSA system (Perkin-Elmer). Images were acquired on the Leica SP8 Microscope using a standard ×20 and ×63 objectives. The collected images were analyzed and processed with the Leica program and Fiji.

For *dpp* transcript quantification, all pictures were treated and analysed with unique parameters. A stack of six z-slices (=9 µm) containing the signal of interest was selected to generate a 'Maximum Intensity Z-projection'. Background was subtracted from the 'Maximum Intensity Z-projection'. A relative signal was obtained by the ratio of mean grey values of 488 channel to mean grey value of the DAPI channel of the region of interest. A relative background was obtained identically using the same ROI outside of the dpp signal. Finally, 'relative signal over background' was obtained from the ratio of 'relative signal' to 'relative background'. All together the calculation can be summarized by the following formula: relative signal/background = mean grey value (Alexa488/DAPI)signal/ (Alexa488/DAPI) background. For Tm1 and Fas2 staining, 'maximum intensity Z-projections' were created using Z-stack of 1.1 µm, and, respectively, 15–25 slides and whole embryos stack using Fiji. Embryos were selected for heterozygous or homozygous genotype according to β-galactosidase expression driven via balancer chromosome (CyO-wg>LacZ, TM6-Dfd>LacZ). Quantifications were performed by blind observation of muscle patterns for 50 embryos per genotype (including heterozygous and homozygous mutants without distinction). Different categories of phenotypes were proposed according to the blind observation performed: strong, medium, mild and normal pattern. Homozygous embryos for balancer chromosomes were not always included as the general shape of the embryos was altered, thus modifying the theoretical percentage of penetrance according to genetic laws. Taking into consideration this parameter, the percentage window of genotype–phenotype of the different fly lines were the following:

1. *Ubx¹,tin³⁴⁶/TM6,Dfd>LacZ, brm²,Ubx¹/TM6-Dfd>lacZ:*

– *Ubx¹,tin³⁴⁶* and *brm²,Ubx¹* homozygous: 25–33% (only strong phenotype).
– *Ubx¹,tin³⁴⁶/TM6,Dfd>LacZ* and *brm²,Ubx¹/TM6,Dfd>LacZ* heterozygous: 50–66% (hardly visible (normal) to mild phenotype).
– *Balancer homozygous (TM6-Dfd>lacZ /TM6,Dfd>LacZ):* 1–25% (normal phenotype or too altered).

2. *Srp54^{DG02112}/CyO-wg>lacZ;Ubx¹/TM6-Dfd>lacZ, snRNPU1-70K^{02107}/CyO-wg>lacZ; Ubx¹/TM6-Dfd>lacZ:*

– Double homozygous *Srp54^{DG02112};Ubx¹* and *snRNPU1-70K^{02107};Ubx¹*: 6.25–11% (strong phenotype).
– *Srp54^{DG02112}/CyO-wg>lacZ;Ubx¹* and *snRNPU1-70K^{02107}/CyO-wg>lacZ;Ubx¹*: 12.5–22% (medium to strong phenotype).
– *Srp54^{DG02112}/CyO-wg>lacZ;Ubx¹/TM6,Dfd>LacZ* and *snRNPU1-70K^{02107}/CyO-wg>lacZ;Ubx¹/TM6,Dfd>LacZ:* 12.5–22% (medium to strong phenotype).
– *Double heterozygous Srp54^{DG02112};Ubx¹/TM6,Dfd>LacZ* and *snRNPU1-70K^{02107};Ubx¹/TM6,Dfd>LacZ:* 25-44% (mild phenotype).
– Single heterozygous: 1–37.5% (normal phenotype).
– Homozygous for balancers: 1–6.25% (normal phenotype or too altered).

Deadpan staining was used for neural cells and neuroblasts quantification and Engrailed for marking the segment boundaries. The numbers of cells per segments were counted, using Z-stack of stage 17 embryos in ventral position.

Fasciclin 2 staining from ×63 focal length was used to analyse motoneurons phenotype by quantification of the innervation of the first ventral-lateral muscle (VL1) of abdominal A2–A7 segments. Phenotypes were classified as followed: normal, misrouted/no connexion, reduced connexion, for which innervation is reaching the muscle but no connexion is observed. Statistical analyses were performed using one-way ANOVA and Chi$^2$ test.

**Mass spectrometry analysis**. Each experiment included nlsGFP, Ubx^{WT} and Ubx^{N51A} samples and was performed in four independent biological replicates. Raw mass spectrometry data were analysed using MaxQuant free software including the Andromeda search Engine[81–83]. Peptide identification was performed using Uniprot database of *Drosophila melanogaster* (canonical and isoform). Default parameters of MaxQuant were used with the following modifications: digestion by Trypsin/P and LysC, lysine biotinylation as variable modification (as well as methionine oxidation and N-terminal acetylation), cytosine carbamido-methylation as fixed modification, Instrument set Orbitrap (with precursor tolerance 20 ppm, MS tolerance 0.5 Da), match between runs option was activated, FDR 1%, label-free quantification (LFQ) and iBAQ calculated (Supplementary Data 1–3). Protein enrichment was calculated using the LFQ Log2 ratio (WT/GFP, N51A/GFP) and normalized on the median value (Supplementary Data 4–33). For each ratio, distribution (90%) and corresponding standard deviations (SD) were calculated to define the proteins significantly enriched (ratio > confidence interval defined as median ± 2 SD). Imputation of value divided by 0 (referred to infinite) has been performed for confidence intervals calculation (Supplementary Data 4–27). Each ratio is then referred as a replicate, related to a list of protein significantly enriched (Supplementary Data 28–34). Subsequently, proteins significantly enriched in at least 2 replicates were considered biologically relevant taking into account biological variability and stochasticity of the MS-process and used for further analysis (Supplementary Data 28–35). Enriched proteins from the different ratio (WT/GFP, N51A/GFP) were then compared with discriminate proteins enriched in the chromatin fraction (WT/GFP, N51A/GFP excluded), from the one enriched more generally in the nucleus (WT/GFP + N51A/GFP) and the one enriched more freely in the nucleoplasm (N51A/GFP, WT excluded) (Supplementary Data 36–38).

**Data analysis and visualisation**. For proteome analysis, Perseus free software was used to generate dot-plot (Pearson, Valid pair value) and clustering visualization (heat map and PCA)[84], based on LFQ log10 value of protein expressed after Perseus canonical filtering (Reverse, Potential Contaminant, Only identified by site) and replacement of missing values.

Functional networks of Ubx interactome were generated with STRING software[85], based on 0.150 interaction score of experimental evidence and database and pathway co-occurrence. Visualization of networks was built with Cytoscape free software[86].

For GO-Term annotations and over-represented GO-Term related to biological process analysis was performed with the web-tools PANTHER using Fisher test and FDR correction. Comparison of Ubx and Tin[45] and Ubx and Grh[46] genomic profile was done as described[2]. The subsequent motif searches on defined regions of 1 kb were performed with the web-tools AME of the MEME-suite with default parameters and fisher test.

Statistical analyses were performed using one-way ANOVA (luciferase assay, signal intensity of mRNA dpp expression level, genetic interaction quantification) and Chi$^2$ test for VL1 innervation phenotype to genotype analysis of motoneuron pattern.

**Reporting summary**. Further information on research design is available in the Nature Research Reporting Summary linked to this article.

**Data availability**
Raw data of MS analysis, Uniprot and contaminant databases and Maxquant files that support the findings of this study have been deposited in PRIDE (https://www.ebi.ac.uk/pride/archive) with the accession code PXD0144818.
Freely accessible datasets used in the study are listed below:
ChIP-on-ChIP of Tin: GSE41628.
ChIP-seq of Grh: GSE83305 using 5–6 h ChIP-seq collection.
Tissue-specific transcriptome and upon Ubx depletion: GSE121670.
Tissue-specific ChIP-seq of Ubx: GSE121752.
The source data underlying Figs. 1–5 and Supplementary Figs. 1, 5–7, 9 are provided as Source Data file. Other raw files are available from the corresponding author upon reasonable request.

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

## Acknowledgements

We thank the Bloomington Center for fly lines, the DHSB for antibodies, and for plasmids the Drosophila Genomics Resource Center, supported by NIH grant 2P40OD010949. We further thank for sharing their materials, Muriel Boube (antibody Med19), Julia Zeitlinger (antibody Zelda), William J. Brook (antibody Cg), Manfred Frasch (Tin antibody, $tin^{346}$ mutant fly line and Tin cDNA construct used for further cloning), David Arnosti (antibody CtBP), Jurg Müller (antibody Pc), Andy Saurin (antibody M1BP), Bill McGinnis (antibody Grh). We warmly thank Ana Rogulja-Ortmann for the dpp674-luciferase construct, *scabrous-GAL4* line, for providing her expertise on the neural system during the revision and the Deadpan antibody (generous gift from Jürgen Knoblich). We are very grateful for all the people who helped to improve the manuscript, in particular Julien Bethune, Guido Grossmann, Jan Lohmann, Justin Crocker, Gislene Pereira and Pedro Pinto, who also provided strong support for microscopy acquisition. This project was supported in part by CellNetworks—Cluster of Excellence (EXC81, J.K.) and DFG LO 844/8-1 (I.L.).

## Author contributions

Conceptualization: J.C. and I.L. Experimental design: J.C., I.L. and J.K. Experimental procedures: J.C., G.S., K.D., S.R. and C.E.P.B. Data analysis: J.C. and G.S. Resources: J.K. Writing: J.C. and I.L. with support of K.D. and G.S. Reviewing: J.C. Funding acquisition: I.L.

## Competing interests

The authors declare no competing interests.
