## [Peer Review File · Nature Communications]

Reviewers' comments:

Reviewer #1 (Remarks to the Author):

The work of Carnesecchi et al. relates to the longstanding issue of tissue-specific activities of Hox proteins. Here authors propose to identify tissue-specific cofactors of the Hox protein Ultrabithorax (Ubx) by establishing the sensitive BioID-based approach in the *Drosophila* embryo.

Analysis from three different tissues (mesoderm, nervous system, neuro-ectoderm) led to the identification of various types of cofactors, ranging from classical transcription factors (TFs) to regulators involved in chromatin remodeling or initiation of translation. This set of results showed that Ubx could act at different levels for gene regulation. In addition, most of these interactions were found to be lineage-specific although they occurred with broadly expressed proteins. This interesting observation suggests that tissue-specificity is not fully relying on the interaction with tissue-specific cofactors. Authors confirmed this hypothesis by doing functional analyses with several new cofactors of Ubx captured in the BioID approach.

Overall this is a massive paper that (i) established a new method for sensitive capture of protein partners in *Drosophila* and (ii) proposed a new model based on robust data for understanding Hox tissue-specificity in vivo. In particular, the role of few tissue-specific cofactors for making a link between Hox proteins and other general cofactors is a very appealing model that could likely apply to other major developmental regulators. In conclusion, both the experimental approach and outreach results make the work of Carnesecchi et al. well suitable for Nature Communication.

Several major and minor points should however be considered by the authors before acceptance for publication.

Major points:

- Because of the huge amount of data it is important to make it more accessible. For example the information on how interactions were considered as biologically relevant (found in two replicates at least) should be provided in the main text or a main figure legend (for instance this information appears only in the M&M section). With this regard what is the proportion of interactions found only once? Are some cofactors found in three or even four replicates? If so, could these interactions be considered as more biologically relevant? What about the nlsBirA* construct: did this construct lead to more unique or more common interactions in replicates when compared to Ubx constructs? In sum, it will be interesting to know whether a ranking of interactions could be done from the BioID purification itself.
- Provide a table with only the cofactors considered as biologically relevant for Ubx and Ubx51.

- One key control protein is Ubx51, which does not bind DNA. Unfortunately, control experiments with Ubx51 are not really convincing. For example, in Supp. Fig.1c, there is clearly less Ubx51 than Ubx in the input, making the interpretation of Exd-coIP difficult. Not surprisingly, it seems that Ubx51 is systematically less biotinylated than Ubx from immunostaining in the embryo. How variable is it? Is Ubx51 systematically less biotinylated than Ubx? Along the same line, authors compared interaction properties of specific cofactors with Ubx and Ubx51. They concluded that CtBP and Tin preferentially interacted with Ubx, but the level of Ubx51 is less than Ubx in the input (Fig. 4a).

- With regard to the validation of interactions in S2 cells, authors should provide a table recapitulating their observations. For example when considering Tin, how many times it was found in the replicates of Ubx and Ubx51, and how strong was the interaction when tested in S2 cells? This should be done systematically for the individual tested cofactors. All westerns could potentially be shown in Supplementary for better clarity in the main figures and Tables. In line with this kind of experiment, testing 2/3 negative cofactors from the BioID purification (never found or found only once) should be considered to further validate the relevance of the approach.

- In the search for enriched binding sites in the mesoderm, the sequence provided for Exd is not convincing (quite degenerate). Why authors did not look more precisely at consensus Hox/Exd binding sites (it will make more sense with regard to the role of Exd)? Are there enriched Hox/Exd binding sites?

- The molecular dissection of Ubx/Tin interaction with several truncated/mutated forms was performed with GST-pull down assays, which are DNA-binding independent. Given the DNA-binding dependency of this interaction, these assays should be repeated in the presence of DNA (classical EMSAs), with a probe derived from either the Dpp enhancer or the consensus Hox/Tin binding site.

Minor points:

- There is a repetition of "Intriguingly". Authors should avoid this term when not necessary (which is the case in several instances).

- The last sentence of the summary is a bit too much over-extrapolated. I suggest to remove or to attenuate this conclusion. First, authors provided evidence that tissue-specific cofactors like Tin or Grh are important for mediating tissue-specific interactions with more ubiquitously expressed proteins. Second, their analysis rather showed that there are tissue-specific TF-enhancer interactions that correlate with tissue-specific expression (especially in the mesoderm).

- Supp. Fig1b: the interaction with Med19 is not really convincing. Authors could focus on the interaction with Exd only (considering the previously mentioned caution with Ubx51).

- The information on partners of Ubx based on their expression profile from previous work (Domsch et al.) should be provided more precisely, with values, in a supplementary table. This information is important given the central message of the paper.

- It would be informative to have the heatmap of Ubx51 and get more easily accessible (digested) information about the common and specific interactions between Ubx and Ubx51 (as a supplementary table). For example, are DNA-binding interactions more lineage-restricted or not?

- Fig. 8 is not really illustrative of the main message of the paper: the information of how Ubx could reach tissue-specificity with general cofactors is missing. It will be more useful to have a final figure focusing on this main message of the paper.

Reviewer #2 (Remarks to the Author):

The article entitled “Multi-level and cell type specificity of protein interactome assembly by a Hox Transcription factor” by Carnesecchi et al., aimed to characterize various interactomes of a nuclear transcription factor, Ubx, in multiple drosophila embryo cell types to elucidate the dependency of various protein interactions on Ubx’s specific function within each specific cell type. The group utilized a cell-type directed in vivo proximity labeling BioID approach coupled with high resolution mass spectrometry to identify and quantitate these various Ubx interactomes and followed up on this discovery work with ample validation of interesting findings.

Overall, this is a well controlled and well thought out experimental design which accounts for non-specific interaction discovery data; in the sense of non-specific protein biotinylation (nucleus directed GFP-BioID as a background control) and functional specificity of Ubx (WT Ubx vs mutant that abolished binding to chromatin). There are, however, several significant concerns about thoroughness of the proteomics discovery data in both how it is presented in the manuscript/supplemental data and the transparency of the resulting protein identification and quantitation metrics.

Comments;

1. The experimental detail included in the “Mass Spectrometry Preparation” sections in Materials and Methods needs to be drastically expanded. Specifically, very little detail is given on how the mass spectrometer was operated (scan settings, resolution, duty cycles, agc settings, etc.) to acquire the data. Unfortunately, in the field of proteomics there is often a lack of transparency on these details which could have profound consequences on experimental replication of these findings in another laboratory.

2. In the same section, please add solvent systems (mobile phase A/B composition; acetonitrile with 0.1% formic acid?) in the description of the LC separation.

3. The description of the analytical UPLC column (Acclaim PepMap RSLC) should not read “75 um x 2um” as its dimensions, but rather 75 um x 50 cm UPLC column packed with 2 um Acclaim PepMap RSLC particles.

4. Although the authors list the proteins identified, there are critical pieces of information not readily accessible to the reader (including supplemental data) including; A) number of unique peptides identified to each protein, B) consistency of expression across biological replicates (easily expressed as intragroup coefficient of variations), C) enrichment values vs background (as fold change or log₂ fold change) or p-value.

5. No AUC or peak intensity quantitative data is presented in the manuscript (including supplemental data). It is nearly impossible to differentiate the interaction confidence of any of the presented interactors with the information presented. No p-values/fold changes nor any type of technical vs biological variation data is presented. The authors did upload MaxQuant output files (zipped files @ 60Gb, much larger unzipped), however it would take significant knowledge of that program, expertise in reading this file structure, and additional software to be able to calculate those values from the raw data uploaded. This is beyond what the scientific community should be responsible to do. Quant values need to be presented in table form for at least those species of interest with associated p-values.

6. Please justify the criteria that proteins significantly enriched in only 2 of the 4 replicates was sufficient to be considered biologically relevant. Depending on that particular proteins intragroup technical+biological variation, this 50% criteria could easily be an artifact from aberrant signals.

7. Please describe the roll up method used for going from the peptide level measurements and area-under-the-curve calculations to the protein level.

8. Please indicate peptide and protein level false discovery rates used within MaxQuant.

Reviewer #3 (Remarks to the Author):

Manuscript NCOMMS-19-20305

Title: "Multi-Level and Cell Type Specificity of Protein Interactome Assembly by a Hox Transcription Factor"

Authors: Julie Carneseccchi, Gianluca Sigismondo, Katrin Domsch, Clara Baader, Mahmoud-Reza Rafiee, Jeroen Krijgsveld, and Ingrid Lohmann

This manuscript addresses the central issue of context-dependency in transcription factor function. This issue is particularly puzzling for the Hox homeotic genes, which act like GPS coordinate cues, instructing all cells in a segment regarding positional identity. This means that Hox genes act in a whole range of tissues/cells within their segments, governing a number of developmental outcomes, unique to each tissue/cell type, and unique to that region. How Hox genes can act in this context-dependent manner is largely unknown. The current study combines the BioID technology, the powerful molecular genetic tools available in *Drosophila*, and the in-depth knowledge of Hox function in the developing embryo, to identify protein interaction partners for the well-studied Hox factor Ubx. By cleverly using both a wild type and a DNA-binding Ubx construct, they can furthermore discriminate between protein interactions occurring on, versus off, the DNA. They also compare the protein interaction landscape for three tissues; mesoderm, neuroectoderm, and nervous system, and intriguingly find that the interaction network differs between the three tissues. They go on to focus on a subset of interaction partners, and confirm the identified protein interactions by other methods, as well as identify mutant phenotypes that are in line with the differential Ubx interactomes.

These findings represent a major leap forward in our understanding of TF context-dependency, and will be of interest to a whole range of scientists interested in developmental biology, gene regulation and context-dependent gene function. However, the study is not without faults, and there are a number of things that could help improve it.

Major issues:

1) The main weakness of the study pertains to Ubx function in the CNS. This refers both to the actual experiments conducted, and to the minimalistic description of Ubx function in the CNS.

First, they only mention one role of Ubx; the control of PCD in subsets of motor neurons. However, they do not actually use this as readout. Instead, they assess motor axon pathfinding (Supplemental Figure 8), but do not describe the previously published phenotype, and only show low-resolution images that are difficult to interpret. Moreover, they claim that there is no motor axon phenotype in Ubx/Ubx, but a recent publication indeed revealed a motor axon projection defect in Ubx/Ubx

mutants (Hessinger et al 2017; they already reference this paper, but for a different purpose). Against this backdrop, it is difficult to evaluate the phenotypes, or lack thereof, of the other genes studied.

Second, other known roles for Ubx in the CNS are very similar to the known roles of Grh, and since they do find Grh as a partner, and go on to specifically study this particular interaction to some extent, these particular roles of Ubx (and Grh) would be worth emphasizing. Specifically, while Hox genes, based upon *in situ* hybridization, do express in the early neuroectoderm, Hox proteins are not initially expressed in NBs (PMIDs 28392108). This is due to the absence of Elav protein, which is only expressed in neurons and stabilizes Ubx mRNA (PMID 24803653). In fact, the lack of Elav protein in glia also explains the lack of Hox protein expression in glia (PMID 15537690), something that can be mitigated by ectopic expression of Elav (PMID 24803653). Moreover, Hox expression is also repressed in NBs by early NB factors (PMID 29112852). Relatedly, Grh is the last gene in the temporal gene cascade (reviewed in PMIDs 24400340, 23962839) and thus expressed late in NBs. Hence, Ubx and Grh become co-expressed late in NBs, as an effect of the aforementioned three regulatory mechanisms. In late NBs they regulate three biological events: the Type I->0 daughter proliferation switch (PMIDs 25073156, 28392108), NB cell cycle exit (PMIDs 9651493, 28392108, 20485487), and NB PCD (PMIDs 28392108, 20485487, 16049114). Predicted direct targets for the first two events are *dap*, *CycE*, *E2f1* and *stg*, while the RHG family (*rpr*, *hid*, *grm*, *sickle*) are likely targets for the PCD. The finding of a physical interaction between Ubx and Grh is very interesting, but there is no functional follow-up regarding what this could mean inside the CNS, in spite of a substantial body of work regarding their similar roles in the CNS.

2) The finding that Ubx mostly interacts with ubiquitously or rather broadly expressed proteins raises the issue of sensitivity. Would Ubx-interacting partners that are expressed only by small sub-sets of cells actually be detected by BioID, and what is the limit? The cell culture system allows them to determine the sensitivity of the method, by mixing in different percentages of mB*Ubx/Exd co-expressing cells with mB*Ubx-only expressing cells (or untransfected control cells), and scoring for biotinylation of Exd by mass spectroscopy. Alternatively, they could drive mB*Ubx with a more restricted driver(s) and assess the BioID profile.

3) Figure 3a: They show the interaction network for Ubx partners in the mesoderm (*twi-Gal4*). It would be important to also show the same networks for *sca-Gal4* and *elav-Gal4*.

4) Figure 5: It would be important to include, in the supplement, gene lists of the genes that are: co-bound by Ubx-Tin and by Ubx-Grh (5a); overlap and differ in Ubx-Tin versus Ubx-Grh binding (5d); overlap in transcriptome versus DNA-binding (5e).

Minor issues:

5) It is a common misconception, but elav-Gal4 is not a pan-neuronal driver. While the Elav protein is only present in neurons, the elav-Gal4 driver drives robustly in neuroblasts, neurons and glia, starting from St11 and onward (PMID 17994541). So it is a pan-neural driver.

6) Page 13: I am surprised that only 18% of genes bound by Ubx-Tin in the mesoderm were affected in Ubx mesodermal mutants? Also, this statement does not refer to any specific figure/table.

7) Figure S2d; missing “myc” labels.

8) Please number the pages and rows.

9) Figure 3d-g' and 4b: I found it difficult to assess overlap. I suggest also showing single immunostained images.

POINT-BY-POINT RESPONSE TO THE REVIEWERS:

We would like to thank all the reviewers for their constructive criticism and important suggestions, which have helped to improve the manuscript significantly. We have included new data and have changed some aspects according to reviewer suggestions, which we describe in detail below.

Reviewer #1: thought the study is well suited for Nature Communication, as it establishes a new method for sensitive capture of protein partners in *Drosophila* and proposes a new model based on robust data for understanding Hox tissue-specificity *in vivo*. In particular, the reviewer thought that the role of few tissue-specific cofactors for making a link between Hox proteins and other general cofactors is a very appealing model that could likely apply to other major developmental regulators.

There were a few concerns raised by this reviewer, which we addressed in the following way:

Major Points:

Because of the huge amount of data, it is important to make it more accessible. For example, the information on how interactions were considered as biologically relevant (found in two replicates at least) should be provided in the main text or a main figure legend (for instance this information appears only in the M&M section). With this regard what is the proportion of interactions found only once? Are some cofactors found in three or even four replicates? If so, could these interactions be considered as more biologically relevant? What about the nlsBirA construct: did this construct lead to more unique or more common interactions in replicates when compared to Ubx constructs? In sum, it will be interesting to know whether a ranking of interactions could be done from the BioID purification itself.*

We thank the reviewer for his interest in the method and to help us providing a more in depth and clearer view of the data.

1) The detail of the calculation was added in the legend of Figure 2 and the detail of the replicates added in the main text.

2) Of note, as developed in the response to reviewer #2, the Supplementary Tables 1, 2, 3 summarizing, for each tissue and replicate, the unique peptide number, LFQ value, the table of the calculation of log₂-LFQ ratio and the summary of the confidence interval calculation (Supplementary Tables 4, 5, 6) are also now included in order to provide a better accessibility to the data processing and pipeline for reader from different fields of research. The lists of the proteins significantly enriched in all replicates are provided in Supplementary Tables 7, 8, 9.

3) The proportion of the proteins/partners found in common between replicates of each tissue are now provided in Supplementary Tables 7, 8, 9 as well as illustrated by Venn diagram for each tissue, WT and N51A Ubx proteins. In addition, we provide now the list and related Venn diagram in Supplementary Table 10 of the protein found in common between each tissue and identified at least in one replicate. This further re-enforces the specificity and unique signature of the tissue-specific interactome but also provide further information for future study on these interaction partners.

4) The reviewer also asked if the interaction found in more than 2 replicates might be more relevant, if it could be used for “ranking” the partners (Supplementary Tables 7, 8, 9). We do not believe that it is the case, as this ranking might be dependent on the inherent variation of protein biotinylation between embryos (that we normalized by pooling), and the intrinsic stochasticity of the MS-process. Another argument is that, interaction and ranking might not be always correlated with unique function. As an example, Tin was enriched in only 2/4

replicates. In contrast, Tin is a strong interactor of Ubx, as revealed by interaction studies (co-IP, GST-pull down) and highly relevant biologically as shown by genetic interaction and functional assay, confirming the difficulty to apply a ranking based on replicate fishing. That is why we chose to rely on a stringent selection based on replicate enrichment combined with the validation with other approach (biochemical and functional). Hypothesis might be proposed for the one enriched in more than 2 replicates, though taking into account that it could be biological variability instead of true partner-ranking.

5) The reviewer further mentioned the nls-BirA* construct. We are actually not sure what the reviewer is asking precisely. But maybe this explanation can clarify the situation. In this study, we did not use a nls-BirA* but the BirA*nls-GFP fusion construct. We used this construct as a general control to discriminate between proteins being biotinylated due to Ubx (as proteins are brought in close proximity to BirA* thanks to Ubx) or whether proteins are biotinylated because they come in close proximity to BirA* for reasons unrelated to Ubx, as all proteins are translated and nuclear proteins are imported into the nucleus etc. Thus, we use BirA*nls-GFP as a general control to identify those that specifically interact with Ubx in the different tissues, due to its size similar to BirA*-Ubx (more than the nls-BirA* alone).

Provide a table with only the cofactors considered as biologically relevant for Ubx and Ubx51.

We thank the reviewer for highlighting this essential dataset. The table was already provided in the word file, it is now added as excel file as Supplementary Table 12 for more transparency.

One key control protein is Ubx51, which does not bind DNA. Unfortunately, control experiments with Ubx51 are not really convincing. For example, in Supp. Fig.1c, there is clearly less Ubx51 than Ubx in the input, making the interpretation of Exd-coIP difficult. Not surprisingly, it seems that Ubx51 is systematically less biotinylated than Ubx from immunostaining in the embryo. How variable is it? Is Ubx51 systematically less biotinylated than Ubx? Along the same line, authors compared interaction properties of specific cofactors with Ubx and Ubx51. They concluded that CtBP and Tin preferentially interacted with Ubx, but the level of Ubx51 is less than Ubx in the input (Fig. 4a).

1) We thank the reviewer for his/her careful reading and highlighting an important point. We would like to stress that the Figure 1c is a BioID (not a co-IP) and the AP fraction is containing biotinylated proteins. The level of proteins is thus due to biotinylation level, meaning the number and accessibility of the lysine residue. For example, GFP is less biotinylated itself, probably due to less lysine residues, less accessible. Thus, we did not compare the BirA* fused protein itself (in term of auto-biotinylation) as it will be different due to the intrinsic nature of the protein. In contrast, BirA* has the same activity, indicating that a close-proximity partner will be biotinylated at the same level if it is in close proximity with the same frequency (and distance). Thus, we can compare the efficiency of biotinylation of a same protein/partner with the different BirA fused proteins (and not the constructs themselves).

2) In line, we performed another BioID-control experiment with adjusted level of BirA*-fused proteins. We also now provided the related quantification of Exd-enrichment (normalized to the expression level given by the input). The AP-ratio of flag-Exd with N51A/WT normalized to the basal expression flag-Exd is N51A:WT=0.3:1. Of note, we observed a slightly slower expression of UbxN51A compared to UbxWT (0.8:1) as the reviewer also noticed. However, the differential expression level of N51A and WT Ubx constructs cannot account for the difference of enrichment of Exd in the AP fraction N51A compared to WT (0.3:1).

- 3) In this way, the Supplementary Figure 1e (BioID1 vs BioID2) confirmed a similar expression level between Ubx N51A and WT.
- 4) This slight differential expression between the constructs are also the reason why we chose to compare the calculation over the control of each of them WT/GFP and N51A/GFP and not WT/N51A, as it might insert a bias due to biotinylation level/protein level. We then compared the presence or absence of partners in the one found at least in two replicate in WT/GFP on a side and N51A/GFP on the other side. We chose not to apply a ranking that might not be relevant biologically taking all these bullet points into account.
- 5) In contrast to the BioID, the coIP experiments may give a better ranking, even though it is based not on frequency and distance but strength & stability of the interaction. In line, co-IP experiments shown in Supplementary Figure 6f, and now quantified in summary Supplementary Table 13, revealed a similar enrichment of Exd with Ubx WT or N51A. It strongly suggests that, more than the interaction-strength, the interaction frequency of Exd-UbxWT is higher than the one of Exd-UbxN51A, opening a large avenue for studying the dynamic of the Exd/Hox complex in the future.

With regard to the validation of interactions in S2 cells, authors should provide a table recapitulating their observations. For example, when considering Tin, how many times it was found in the replicates of Ubx and Ubx51, and how strong was the interaction when tested in S2 cells? This should be done systematically for the individual tested cofactors. All westerns could potentially be shown in Supplementary for better clarity in the main figures and Tables. In line with this kind of experiment, testing 2/3 negative cofactors from the BioID purification (never found or found only once) should be considered to further validate the relevance of the approach.

We thank the reviewer for her/his thoughtful advices for improving the clarity of the manuscript. As requested, we provided now a more comprehensive table in Supplementary Table 13 containing the different partners, positive and negative controls tested. We feel that the provided table clearly shows that we already tested a set of positive (Exd, Med19, M1BP) and negative (Ncm, Tubulin, Pc) control, whether in cells or at the endogenous level that validate the whole approach. Moreover, as suggested, coIP-related immunoblots are now grouped in the Supplementary Figure 6.

In the search for enriched binding sites in the mesoderm, the sequence provided for Exd is not convincing (quite degenerate). Why authors did not look more precisely at consensus Hox/Exd binding sites (it will make more sense with regard to the role of Exd)? Are there enriched Hox/Exd binding sites?

We deeply appreciate the reviewer's interest and careful reading of the different analyses performed in the study, from BioID to genomic analysis. Indeed, the Exd binding site provided is quite degenerate, as it is the result of an unbiased search of enriched motifs using MEME software. From MEME, as mentioned, the search is not biased. Thus, there is not a directed search specifically for Ubx/Exd and if the site is not present, it is just revealing that it is not enriched primarily. However, it also intrigued us; therefore, we searched for enrichment of Hox/Exd high affinity site using the Selex-seq matrix provided by the work of the lab of Richard Mann:

dataset	adj_p-value
Meso-Ubx	3.87E-02
Ubx-Tin	1.00E+00

Neuro-Ubx 1.00E+00
Ubx-Grh 1.00E+00

The analysis revealed a significant enrichment of the Hox/Exd sites only in the mesoderm-specific ChIP-seq of Ubx (and the p-value is rather low). None of the common Ubx-Tin, Ubx-Grh as well as neural-specific ChIP-seq of Ubx is enriched for this optimal Hox/Exd binding site. It opens a diversity of hypothesis concerning: 1, the requirement of Exd/Hox complex to regulate these target genes; 2, the biological requirement of this optimal site for driving Ubx transcriptional function *in vivo*.

The molecular dissection of Ubx/Tin interaction with several truncated/mutated forms was performed with GST-pull down assays, which are DNA-binding independent. Given the DNA-binding dependency of this interaction, these assays should be repeated in the presence of DNA (classical EMSAs), with a probe derived from either the Dpp enhancer or the consensus Hox/Tin binding site.

1) We thank the reviewer for her/his interest to the molecular dissection of Ubx/Tin interaction. Importantly, we clarified that the interaction is not DNA dependent (interaction *in vitro* without DNA) but the functional cooperation is (Luciferase assay, Genetic interaction). The question of DNA-binding cooperation is indeed relevant and we now provide as Supplementary Figure 7e the EMSA on the dpp enhancer.

This experiment re-enforces the fact that: 1), they both directly bind the dpp enhancer, in a range of 100bp, 2), they are able to bind both in complex illustrated by the supershift of the 'Ubx+Tin' band +V5/MBP antibody, (most probably too heavy to enter the gel), and independently, illustrated by the single binding.

2) Of note, the first data did not show a DNA-dependency of the interaction as GST-pull down experiments showed a direct interaction between Ubx and Tin without the need of DNA. In contrast, colIP and BioID experiment revealed that, in a more physiological context, the interaction is happening more often (BioID, frequency) and is probably more stable (colIP, N51AvsWT Ubx) on DNA (see now summary of colIP quantification, in Supplementary Table 13). This could explain why the results of the EMSA experiments are difficult to interpret in term of cooperation at the DNA-binding level. In order to decipher the requirement of DNA-binding abilities of both Tin and Ubx, we are now providing luciferase assay with Tin WT or N51A (Supplementary Figure 7b), showing that a functional DNA-binding domain of both Tin and Ubx is required for functional cooperation on the dpp enhancer (bound by ChIP for Ubx and Tin, also shown now by EMSA). To conclude, it strongly suggests that both interaction and functional DNA binding are required for proper transcriptional cooperation.

3) The question of the cooperation for DNA-binding/chromatin-loading might be explored in future, using more sensitive *in vivo* method such as Fluorescence after photobleaching to study the residence time on the chromatin and thereby the influence on each other DNA-interaction (strength, frequency). We already have preliminary data showing that, indeed, interaction (and not DNA-binding) is sufficient for enhancing Tin chromatin residency by Ubx (in *Drosophila* S2R+ cells). we are happy to share this result with the reviewer, but we feel that it is going beyond the scope of this manuscript and will only be presented here as personal communication.

Recovery after photobleaching of GFP-Tin

Minor Points:

There is a repetition of “Intriguingly”. Authors should avoid this term when not necessary (which is the case in several instances).

The text has been modified accordingly.

The last sentence of the summary is a bit too much over-extrapolated. I suggest to remove or to attenuate this conclusion. First, authors provided evidence that tissue-specific cofactors like Tin or Grh are important for mediating tissue-specific interactions with more ubiquitously expressed proteins. Second, their analysis rather showed that there are tissue-specific TF-enhancer interactions that correlate with tissue-specific expression (especially in the mesoderm).

“Taken together, our work reveals that TF-interaction networks underlying cell type specification may be much more complex than previously thought and challenges two long-standing assumptions in developmental biology, namely the relevance of differential RNA expression as determinant for protein complexes and the focused view on TF-enhancer interactions for lineage specific gene expression.”

We feel that the sentence is already attenuated as we precised it “may [...] challenge”. As the abstract has been reduced do to editorial policy of Nature Communication, the sentence has been in addition shortened, which we hope will satisfy the reviewer.

Supp. Fig1b: the interaction with Med19 is not really convincing. Authors could focus on the interaction with Exd only (considering the previously mentioned caution with Ubx51).

We feel that the Med19 AP-fishing is worth it to keep. Though, the level is low, it is the validation of the method for capturing a partner expressed at the endogenous level, in contrast with Exd that is over-expressed. Exd fishing with BioID has been adjusted by new cleaner BioID experiment with BirA⁺-GFP, -UbxN51A and -UbxWT and quantification.

The information on partners of Ubx based on their expression profile from previous work (Domsch et al.) should be provided more precisely, with values, in a supplementary table. This information is important given the central message of the paper.

The data has been now provided in Supplementary Table 11.

It would be informative to have the heatmap of Ubx51 and get more easily accessible (digested) information about the common and specific interactions between Ubx and Ubx51 (as a supplementary table). For example, are DNA-binding interactions more lineage-restricted or not?

The tables of the protein enriched in the UbxN51A, WT or common were already provided but we acknowledge that the word files were not clearly readable. We now provided the tables as Supplementary Tables 7 8, 9 and 10, for each tissue, and presenting the protein enriched in the different fraction (chromatin, nucleus, nucleoplasm) according to UbxWT and N51A partner-enrichment (Supplementary Table 11). We also provided within these files a short information concerning the function or the family of the protein/partners identified. We believe that it will be much better readable and the data more accessible this way.

Fig. 8 is not really illustrative of the main message of the paper: the information of how Ubx could reach tissue-specificity with general cofactors is missing. It will be more useful to have a final figure focusing on this main message of the paper.

We clarified the legend to make it closer to the point.

Reviewer #2: highlighted that this study is a well-controlled and well thought out design.

There were a few concerns raised by this reviewer, which we addressed in the following way:

Major Points:

The experimental detail included in the “Mass Spectrometry Preparation” sections in Materials and Methods needs to be drastically expanded. Specifically, very little detail is given on how the mass spectrometer was operated (scan settings, resolution, duty cycles, agc settings, etc.) to acquire the data. Unfortunately, in the field of proteomics there is often a lack of transparency on these details which could have profound consequences on experimental replication of these findings in another laboratory.

We thank the reviewer to help us improving the clarity and transparency of the MS-processing. The text has been modified accordingly in the Material and Methods.

In the same section, please add solvent systems (mobile phase A/B composition; acetonitrile with 0.1% formic acid?) in the description of the LC separation.

The text has been modified accordingly.

The description of the analytical UPLC column (Acclaim PepMap RSLC) should not read “75 um x 2um” as its dimensions, but rather 75 um x 50 cm UPLC column packed with 2 um Acclaim PepMap RSLC particles.

The text has been modified accordingly.

Although the authors list the proteins identified, there are critical pieces of information not readily accessible to the reader (including supplemental data) including; A) number of unique peptides identified to each protein, B) consistency of expression across biological replicates (easily expressed as intragroup coefficient of variations), C) enrichment values vs background (as fold change or log2 fold change) or p-value.

We acknowledge the reviewer that we did not provide a comprehensive table of the data. We now provide several new files, illustrating the different step of analysis as well as, as described for reviewer #1 answer, more detailed tables of the protein per replicate for each sample. The Supplementary Tables 1, 2 and 3 summarize, for each tissue and replicate, the unique peptide number, LFQ value. The table of the calculation of log2-LFQ ratio and the summary of the confidence interval calculation (Supplementary Tables 4, 5 and 6 and summarized lists in Supplementary Tables 7, 8, 9) are now included in order to provide a better accessibility of the data processing to any reader. Consistency across replicate is represented by Pearson correlation and heatmap (Figure 1 and Supplementary Figures 3, 4).

No AUC or peak intensity quantitative data is presented in the manuscript (including supplemental data). It is nearly impossible to differentiate the interaction confidence of any of the presented interactors with the information presented. No pvalues/fold changes nor any type of technical vs biological variation data is presented. The authors did upload MaxQuant output files (zipped files @ 60Gb, much larger unzipped), however it would take significant knowledge of that program, expertise in reading this file structure, and additional software to be able to calculate those values from the raw data uploaded. This is beyond what the scientific community should be responsible to do. Quant values need to be presented in table form for at least those species of interest with associated p-values.

As mentioned above, the Supplementary Tables 1, 2 and 3 provide the unique peptide number, LFQ value, the table of the calculation of log2-LFQ ratio and the summary of the confidence interval calculation (Supplementary Tables 4, 5 and 6) are now included in order to provide a better data accessibility. Files used to generate Figures 1-3 and Supplementary Figures 3, 4 and Tables 1-9 are reported in the separate quantification files section provided during data submission (15MB in .zip). Quantification values for all the experiments files can be found under the path: "Proteingroups_Carnesecchi et al Lineage-specific protein interactomes of Ubx.zip\Maxquant file - Copy\used for file\".

Please justify the criteria that proteins significantly enriched in only 2 of the 4 replicates was sufficient to be considered biologically relevant. Depending on that particular proteins intragroup technical+biological variation, this 50% criteria could easily be an artifact from aberrant signals.

This point has been stressed in the response of a question of reviewer #1, asking if the replicates could be used for "ranking" the partners. We believe that this ranking/fishing is dependent of the inherent variation of protein biotinylation between embryos, and the intrinsic stochasticity of the MS-process. Furthermore, ranking might not be always correlated with unique function, as exemplified by the strong partner Tin, enriched in 2/4 replicates. We feel that this answer is correlated with the question of reviewer #2 as it is part of the justification of our analysis and selection of the partners. In fact, the intrinsic variation of biotinylation *in vivo*, the purification and inherent stochastic parameter of the MS-process are the major driver of our analysis design. We chose to rely on a stringent selection based: 1. Statistically relevant enrichment ratio of log2-LFQ Ubx/GFP for each independent replicate (Affinity-purified

sample), 2., followed by the selection of the one found significantly enriched at least in 2/4 replicates, 3., combined with the validation with other approach (biochemical and functional). The point 1. is, we believe, a standard set up of calculation using LFQ data. We chose next the selection of 2/4 replicates, that provide the balance between biological variability (we may lose interesting partners like Tin with a selection of 3/4) and false-positive that may appear if partners found at least in one replicate were selected. Further validation by other approaches (interaction and function, point 3.) are re-enforcing the suitability of our pipeline. Though taken carefully, the comparison of partner enriched at least in 1 replicate are now provided in Supplementary Table 10 and might be the primary support for future study. The description of the pipeline of analysis has also been extended in the Material and Methods.

Please describe the roll up method used for going from the peptide level measurements and area-under-the-curve calculations to the protein level.

The peptide level measurement is function of the Andromeda search engine based on the Maxquant analysis. This is a calculation in-built to the engine and provided by the Cox's laboratory. This information has been already added in the Material & Method with the associated publications.

Cox J., Hein M. Y., Luber C. A., Paron I., Nagaraj N., and Mann M., Accurate Proteome-wide Label-free Quantification by Delayed Normalization and Maximal Peptide Ratio Extraction, Termed MaxLFQ. *Mol Cell Proteomics*, 2014, 13, pp 2513–2526

Please indicate peptide and protein level false discovery rates used within MaxQuant.

This information (FDR 1%) has been added in the Material & Method.

Reviewer #3: thought that the findings of this study represent a major leap forward in our understanding of TF context-dependency, and will be of interest to a whole range of scientists interested in developmental biology, gene regulation and context-dependent gene function.

There were a few concerns raised by this reviewer, which we addressed in the following way:

Major Points:

The main weakness of the study pertains to Ubx function in the CNS. This refers both to the actual experiments conducted, and to the minimalistic description of Ubx function in the CNS. First, they only mention one role of Ubx; the control of PCD in subsets of motor neurons. However, they do not actually use this as readout. Instead, they assess motor axon pathfinding (Supplemental Figure 8), but do not describe the previously published phenotype, and only show low-resolution images that are difficult to interpret. Moreover, they claim that there is no motor axon phenotype in Ubx/Ubx, but a recent publication indeed revealed a motor axon projection defect in Ubx/Ubx mutants (Hessinger et al 2017; they already reference this paper, but for a different purpose). Against this backdrop, it is difficult to evaluate the phenotypes, or lack thereof, of the other genes studied.

Second, other known roles for Ubx in the CNS are very similar to the known roles of Grh, and since they do find Grh as a partner, and go on to specifically study this particular interaction to some extent, these particular roles of Ubx (and Grh) would be worth emphasizing. Specifically, while Hox genes, based upon in situ hybridization, do express in the early neuroectoderm, Hox proteins are not initially expressed in NBs (PMIDs 28392108). This is

due to the absence of Elav protein, which is only expressed in neurons and stabilizes Ubx mRNA (PMID 24803653). In fact, the lack of Elav protein in glia also explains the lack of Hox protein expression in glia (PMID 15537690), something that can be mitigated by ectopic expression of Elav (PMID 24803653). Moreover, Hox expression is also repressed in NBs by early NB factors (PMID 29112852). Relatedly, Grh is the last gene in the temporal gene cascade (reviewed in PMIDs 24400340, 23962839) and thus expressed late in NBs. Hence, Ubx and Grh become co-expressed late in NBs, as an effect of the aforementioned three regulatory mechanisms. In late NBs they regulate three biological events: the Type I->0 daughter proliferation switch (PMIDs 25073156, 28392108), NB cell cycle exit (PMIDs 9651493, 28392108, 20485487), and NB PCD (PMIDs 28392108, 20485487, 16049114). Predicted direct targets for the first two events are dap, CycE, E2f1 and stg, while the RHG family (rpr, hid, grm, sickle) are likely targets for the PCD. The finding of a physical interaction between Ubx and Grh is very interesting, but there is no functional follow-up regarding what this could mean inside the CNS, in spite of a substantial body of work regarding their similar roles in the CNS.

1) We thank the reviewer to help us improving the scientific content of the manuscript and acknowledge that the study was lacking more in-depth analysis on the neural system. We strongly thank the reviewer for her/his guidance and already stress that, due to editorial policies, we could not be able to cite all the proposed bibliography that was, undoubtedly, a solid base of the revision process.

Concerning the Fasciclin2 staining, we acknowledged that Ubx mutant phenotype was not detailed in the legend and is now provided. As described in Hessinger et al (2017), the penetrance of the phenotype is rather low. In contrast, the homozygous double mutant of Brm and Ubx, for which we observed an absence of the axon projection in A1 segment, is constant. Moreover, the absence of phenotype associated with the double homozygous mutant of Ubx and the other partners can re-enforce the absence of genetic interaction in this lineage, even if both copy of the 2 partners are absent.

To consolidate the crucial point of lineage-specific functional cooperation, we took advantage of the strong phenotype observed on neuroblast (NB) number in Ubx homozygous mutants, for which we observed and described now a functional cooperation between Ubx and Grh only in the neural system but not in the mesoderm (Supplementary Figure 9). Subsequently, we asked whether interaction partners identified in the mesoderm (Srp54, snRNPU1-70K, Brm, Tin), could affect the NBs number and if double heterozygous mutant embryos present a phenotype using the NBs number as read-out (Supplementary Figure 10). Importantly, this was not the case for most of them, with the exception of Brm, which we also identified by BioID of Ubx in the neural lineage. The new data is now provided in Supplementary Figure 10 and quantified in Supplementary Table 17.

The absence of genetic interaction (except Brm) might still be dependent on the specific phenotype analysed, nevertheless, we feel that 1., the more in depth study provided on neuroblasts number, and 2., the strong phenotype observed for snRNPU1-70K homozygous mutant (in contrast to the absence of phenotype in the double heterozygous mutant) consolidate the weakness of the first version on the manuscript. Moreover, it opens new hypothesis concerning the suitability of the BioID for identifying lineage-specific interactomes, the role of Ubx-Grh complex in neurogenesis but also the role of Ubx-Brm complex on neuroblasts (NBs) type I. So far, Brm function has been linked to NBs type II only, (Koe et al., 2014, elife, PMID: 24618901, Zhang et al., 2019, cell Reports, PMID: 31018143, that we cannot cite due to reference number of editorial policy) which are localized in brain lobe (Alvarez et al., 2018, PMID: 29567672, Walsh et al. 2017, PMID: 29158446). Studies on the Brm-Ubx transcriptional and functional cooperation in neurogenesis will be of great interest in the future.

2) In line with the guidance of the reviewer on the role of Hox and Grh on neurogenesis, we analysed more in depth the genome wide binding profile and provided the result here as a personal communication. The cooperation between Ubx and Grh in regulating neuroblast number may not depend on the regulation of cell death, as apoptosis-related genes such as reaper, hid, grm and sickle are not commonly bound by Ubx and Grh (Supplementary Table 15). Several genes involved in cell cycle regulation (dap, e2f1 and stg) were co-bound, however, proliferation-related GO-terms were absent from the Ubx-Grh bound genes dataset (Figure 4a, Supplementary Tables 14-15). It is quite interesting to see that cell death seems to not be regulated by the cooperation between Ubx and Grh, while some genes involved in cell cycle regulation are bound by both TFs. Though, this is only correlating genome binding profile, beyond the scope of this manuscript, it provides new entry point for deciphering the functional cooperation if Ubx-Grh in future, and more generally the collaboration between Hox-Grh and other lineage-restricted TF in neurogenesis.

*The finding that Ubx mostly interacts with ubiquitously or rather broadly expressed proteins raises the issue of sensitivity. Would Ubx-interacting partners that are expressed only by small sub-sets of cells actually be detected by BioID, and what is the limit? The cell culture system allows them to determine the sensitivity of the method, by mixing in different percentages of mB*Ubx/Exd co-expressing cells with mB*Ubx-only expressing cells (or untransfected control cells), and scoring for biotinylation of Exd by mass spectroscopy. Alternatively, they could drive mB*Ubx with a more restricted driver(s) and assess the BioID profile.*

1) The reviewer raised an important point concerning the identification of rather ubiquitous proteins. However, we do think that the data are providing the confidence and strength in term of sensitivity. First, the comparison of BioID and tissue-specific transcriptome (Supplementary Table 11) revealed that some interactors such as the mesodermal partner Srp54, are expressed at the same RNA level in the mesoderm and nervous system, some like tfAP-2 (and Grh) are expressed at very low mRNA level (2-10RPKMs) but still detected. Another example is the neural-partner top1 which is expressed (at least at the RNA level) at higher level in the mesodermal-transcriptome than the neural one, confirming that the method is not limited by sensitivity but rather interaction specificity. Second, a perfect example is the fishing of Grh in the neural system, also expressed at very low mRNA level. As exhibited in Supplementary Figure 6g, showing the endogenous co-localisation of Ubx, Grh and elav, a rather small subset of cells express the 3 proteins at these stages, indicating that the elav-BioID captured rare co-occurrences with respect to cell number but still occurring with high frequency within these cells. In contrast, the well-known Exd partner of Ubx, ubiquitously expressed was not captured by the BioID, re-enforcing the suitability of the method to capture sensitive (and more frequent or close-proximity) interactions.

2) Finally, this point is also now illustrated by comparing the BioID and coIP capture of Exd with Ubx WT and N51A. As described in the response to reviewer #1, Exd interaction strength or stability is similar for Ubx WT or N51A as indicated by co-IP. In contrast, BioID, based on interaction frequency and distance, revealed a stronger biotinylation and subsequent affinity-purification of Exd with Ubx WT than the mutant. It strongly suggests that frequency/close-proximity on DNA, more than interaction-strength, is an essential parameter for the functional Exd/Hox complex and further re-enforces the suitability of BioID for deciphering *in vivo* regulatory networks.

Figure 3a: They show the interaction network for Ubx partners in the mesoderm (twi-Gal4). It would be important to also show the same networks for sca-Gal4 and elav-Gal4.

The interaction networks of Ubx partners in the neural and neuroectodermal tissues are now been provided in Supplementary Figures 5c-5d.

Figure 5: It would be important to include, in the supplement, gene lists of the genes that are: co-bound by Ubx-Tin and by Ubx-Grh (5a); overlap and differ in Ubx-Tin versus Ubx-Grh binding (5d); overlap in transcriptome versus DNA-binding (5e).

Of note, Figure 5 in now Figure 4, according to reviewer #1 comments. The corresponding gene lists has been added in the following tables:

Supplementary Table 15: List of genes illustrating the Figures 4a-4e

Supplementary Table 16: List of genes bound by Ubx-Tin and altered in mesodermal knock-down of Ubx using degradFP system from Figure 4

Minor Points:

It is a common misconception, but elav-Gal4 is not a pan-neuronal driver. While the Elav protein is only present in neurons, the elav-Gal4 driver drives robustly in neuroblasts, neurons and glia, starting from St11 and onward (PMID 17994541). So it is a pan-neural driver.

We thank the reviewer to help us improving the accuracy of the scientific vocabulary of the manuscript. The text has been modified has proposed for elav reference as a pan-neural driver and neural tissue.

Page 13: I am surprised that only 18% of genes bound by Ubx-Tin in the mesoderm were affected in Ubx mesodermal mutants? Also, this statement does not refer to any specific figure/table.

The related list of genes has been provided in Supplementary Table 16. Moreover, we acknowledge a mistake in the calculation as 74/367 are affected (so 20% and not 18%) by the Ubx mesodermal Knock Down (KD). This dataset is from Domsch et al, 2019 in which, we performed mesoderm-specific depletion of endogenous GFP-Ubx using the DegradFP system (degradation tissue-specific of the protein via proteasomal system and nanobodies targeting the GFP). In this publication, we hypothesized concerning the rather small amount of genes mis-regulated upon Ubx-KD. In fact, Ubx-KD is associated with an over-expression of AbdA and Antp that might compensate for Ubx absence on a set of target genes. Consequently, the number of target genes bound by Ubx-Tin and affected by Ubx-KD is limited intrinsically by the dataset.

Figure S2d; missing "myc" labels.

The missing tag "myc" has been added.

Please number the pages and rows.

The numbers have been provided

Figure 3d-g' and 4b: I found it difficult to assess overlap. I suggest also showing single immunostained images.

We feel that the images are focused on the co-localisation, the central message driven by the experiments and illustrated by the quality of zoomed pictures. Grh and Ubx co-staining are already presented with and without elav for more clarity. For the mesodermal tissue, the $\text{twi} > \text{RAN}^{\Delta}$ construct is not overlapping the co-localisation of Ubx with the different partners tested as it is localized on the nuclear border. Thus, we are not convinced that adding the single images in the figure would provide more information. Thought, we would be pleased to share as personal communication, the pictures with the reviewer and discuss further hypothesis/question that it might highlight.

REVIEWERS' COMMENTS:

Reviewer #1 (Remarks to the Author):

Authors have satisfactorily replied to all my concerns. In particular, data are more easily accessible and requested experiments, in particular for analyzing the importance of DNA-binding for Hox/Tin complex formation, have been thoroughly performed.

Overall data are presented in a more comprehensive way and the new version of the manuscript gained in visibility and impact.

Reviewer #2 (Remarks to the Author):

This reviewer thanks the authors for addressing initial concerns, mainly focused on technical and availability of proteomics data and methods used to acquire them. All of the concerns that were raised have been adequately addressed, either through additional supplemental tables, figures, or text in the manuscript.

Reviewer #3 (Remarks to the Author):

Manuscript NCOMMS-19-20305A

Title: "Multi-Level and Cell Type Specificity of Protein Interactome Assembly by a Hox Transcription Factor"

Authors: Julie Carneseccchi, Gianluca Sigismondo, Katrin Domsch, Clara Baader, Mahmoud-Reza Rafiee, Jeroen Krijgsveld, and Ingrid Lohmann

Comments on re-submitted manuscript:

The authors have addressed several of the issues I raised, and the manuscript has improved. However, I am still not satisfied with their response to my original point 1. Specifically:

1) Regarding the role of Ubx in motor neurons, they initially claimed that there was no motor axon phenotype in Ubx/Ubx. But now they claim that there is a phenotype. But the figure looks identical to the initial submission. Moreover, the precise phenotype is not clearly described or depicted in the figure: I am not sure what “loss of the characteristic T-shape of the axonal projection” means, it is not a common term used in the motor axon pathfinding field. Finally, there is no quantification of any motor neuron phenotypes, which makes it difficult to assess their statements on rows 439-441 regarding lack of genetic interaction in the CNS between Ubx and its muscle-specific partners. Please a) depict the motor neuron phenotype clearly in the figure, b) describe it more precisely in the figure legend and main text, and c) quantify this phenotype.

Regarding the interplay between Ubx and Grh, it is reassuring to see a genetic interaction with respect to NB numbers. However, the data in Supplemental Table 17 should be provided as raw numbers, not mean, and with the statistical analysis clearly described. Moreover, the NB number phenotype is a PCD effect. Ubx and Grh has no role in NB generation in A1 or A2, but are involved in NB removal by PCD, at the end of lineage development, during stages 13-17, when the majority of NBs in A1 are removed by PCD. Please ensure that the main text and figure legends clearly refers to this phenotype as PCD and not neurogenesis, which is usually interpreted as NB generation (albeit the term neurogenesis is rather imprecise). Also, please ensure that at least some reference is made to the previous studies identifying reduced PCD of NBs in Ubx and Grh mutants (e.g., PMIDs in my original review).

Answer point by point to the reviewer

Reviewer #1 (Remarks to the Author): Authors have satisfactorily replied to all my concerns. In particular, data are more easily accessible and requested experiments, in particular for analyzing the importance of DNA-binding for Hox/Tin complex formation, have been thoroughly performed. Overall data are presented in a more comprehensive way and the new version of the manuscript gained in visibility and impact.

-We thank reviewer #1 for its comment.

Reviewer #2 (Remarks to the Author): This reviewer thanks the authors for addressing initial concerns, mainly focused on technical and availability of proteomics data and methods used to acquire them. All of the concerns that were raised have been adequately addressed, either through additional supplemental tables, figures, or text in the manuscript.

-We thank reviewer #2 for his/her comment.

Reviewer #3 (Remarks to the Author): Manuscript NCOMMS-19-20305A Title: "Multi-Level and Cell Type Specificity of Protein Interactome Assembly by a Hox Transcription Factor" Authors: Julie Carnesecchi, Gianluca Sigismondo, Katrin Domsch, Clara Baader, Mahmoud-Reza Rafiee, Jeroen Krijgsveld, and Ingrid Lohmann Comments on re-submitted manuscript: The authors have addressed several of the issues I raised, and the manuscript has improved.

-We thank reviewer #3 for his/her notice.

However, I am still not satisfied with their response to my original point 1. Specifically:

1) Regarding the role of Ubx in motor neurons, they initially claimed that there was no motor axon phenotype in Ubx/Ubx. But now they claim that there is a phenotype. But the figure looks identical to the initial submission. Moreover, the precise phenotype is not clearly described or depicted in the figure: I am not sure what "loss of the characteristic T-shape of the axonal projection" means, it is not a common term used in the motor axon pathfinding field. Finally, there is no quantification of any motor neuron phenotypes, which makes it difficult to assess their statements on rows 439-441 regarding lack of genetic interaction in the CNS between Ubx and its muscle-specific partners. Please a) depict the motor neuron phenotype clearly in the figure, b) describe it more precisely in the figure legend and main text, and c) quantify this phenotype.

-As requested, we improved the description and analysis of the motoneurons phenotype. In particular, as request, we present now a), zoomed picture decrypting the phenotype for each of the genotype presented. We further highlighted the innervation of the ventral lateral muscle 1 with arrows for each genotype further quantified. b) This specific innervation of the VL1 muscle is affected in Ubx mutant embryos as described now in the legend and the main text. We further c) quantify the genotype of the genetic interaction for single and double heterozygous as well as Ubx mutant and provide statistical analysis using Chi2 test. This showed a significant misrouting/loss of connexion of the innervation of the VL1 muscle in A2-A7 hemisegments (n=30) for Ubx homozygous mutant in comparison to heterozygous mutants, while double heterozygous mutants (Ubx combined with other factors) do not show a significant alteration of the innervation of the VL1 muscle compared to single heterozygous mutants. The raw data are also presented in the source file.

Regarding the interplay between Ubx and Grh, it is reassuring to see a genetic interaction with respect to NB numbers. However, the data in Supplemental Table 17 should be provided as raw numbers, not mean, and with the statistical analysis clearly described.

-As requested, improvements have been done for the genetic interaction relative to NB number. The graphic is now presented as diagram containing dot-plot +-sem for better clarity of the quantification. The raw data of NB counting are now provided within the source file and the statistical analysis is added for the relevant phenotype/genotype.

Moreover, graphic views containing dot-plot +-sem are now provided for the other genetic interaction analysed as well as statistical test (supplementary figure 10). The raw files are also available as table in the source data file with extended details on statistic.

Moreover, the NB number phenotype is a PCD effect. Ubx and Grh has no role in NB generation in A1 or A2, but are involved in NB removal by PCD, at the end of lineage development, during stages 13-17, when the majority of NBs in A1 are removed by PCD. Please ensure that the main text and figure legends clearly refers to this phenotype as PCD and not neurogenesis, which is usually interpreted as NB generation (albeit the term neurogenesis is rather imprecise). Also, please ensure that at least some reference is made to the previous studies identifying reduced PCD of NBs in Ubx and Grh mutants (e.g., PMIDs in my original review).

-As requested, the legend and text illustrating the phenotype are modified, and it is indicated that the effect is due to PCD.

-As requested, the 3 references that were listed in the original review are now cited in the text and legend that referred to the PCD:

PMID 28392108: Anterior-Posterior Gradient in Neural Stem and Daughter Cell Proliferation Governed by Spatial and Temporal Hox Control. Monedero Cobeta I, Salmani BY, Thor S

PMID 20485487: Segment-specific neuronal subtype specification by the integration of anteroposterior and temporal cues. Karlsson D, Baumgardt M, Thor S.

PMID 16049114: Drosophila Grainyhead specifies late programmes of neural proliferation by regulating the mitotic activity and Hox-dependent apoptosis of neuroblasts. Cenci C, Gould AP.